# Integrating Intermediate Layer Optimization and Projected Gradient Descent for Solving Inverse Problems with Diffusion Models

Yang Zheng [1]  Wen Li [1]  Zhaoqiang Liu [1]

## Abstract

Inverse problems (IPs) involve reconstructing signals from noisy observations. Recently, diffusion models (DMs) have emerged as a powerful framework for solving IPs, achieving remarkable reconstruction performance. However, existing DM-based methods frequently encounter issues such as heavy computational demands and suboptimal convergence. In this work, building upon the idea of the recent work DMPlug (Wang et al., 2024), we propose two novel methods, DMILO and DMILO-PGD, to address these challenges. Our first method, DMILO, employs intermediate layer optimization (ILO) to alleviate the memory burden inherent in DMPlug. Additionally, by introducing sparse deviations, we expand the range of DMs, enabling the exploration of underlying signals that may lie outside the range of the diffusion model. We further propose DMILO-PGD, which integrates ILO with projected gradient descent (PGD), thereby reducing the risk of suboptimal convergence. We provide an intuitive theoretical analysis of our approaches under appropriate conditions and validate their superiority through extensive experiments on diverse image datasets, encompassing both linear and nonlinear IPs. Our results demonstrate significant performance gains over state-of-the-art methods, highlighting the effectiveness of DMILO and DMILO-PGD in addressing common challenges in DM-based IP solvers.

## 1. Introduction

Inverse problems (IPs) represent a broad class of challenges focused on reconstructing degraded signals, particularly images, from noisy observations. These problems are of significant importance in a wide range of real-world applications, such as medical imaging (Chung & Ye, 2022; Song et al., 2022), compressed sensing (Wu et al., 2019; Candès & Tao, 2005), and remote sensing (Twomey, 2019). Mathematically, the objective of an IP is to recover an unknown signal $\boldsymbol{x}^* \in \mathbb{R}^n$ from observed data $\boldsymbol{y} \in \mathbb{R}^m$, typically modeled as (Foucart & Rauhut, 2013; Saharia et al., 2022a):

$$\boldsymbol{y} = \mathcal{A}(\boldsymbol{x}^*) + \boldsymbol{\epsilon}, \tag{1}$$

where $\mathcal{A}(\cdot)$ denotes the given forward operator (e.g., a linear transformation, convolution, or subsampling), and $\boldsymbol{\epsilon} \in \mathbb{R}^m$ represents stochastic noise. A key challenge in solving IPs arises from their inherent ill-posedness: When $m < n$, even in the absence of noise, $\boldsymbol{x}^*$ cannot be uniquely determined from $\boldsymbol{y}$ and $\mathcal{A}(\cdot)$. This underdetermined nature necessitates the incorporation of prior knowledge or regularization to ensure stable and reliable solutions.

Traditional approaches for solving IPs often rely on hand-crafted priors based on domain knowledge, such as sparsity in specific domains (e.g., Fourier or wavelet) (Tibshirani, 1996; Bickel et al., 2009). While effective in certain scenarios, these methods are limited in their ability to capture the rich and complex structures of natural signals and are prone to suboptimal performance in highly ill-posed problems.

The advent of generative models has introduced new paradigms for addressing IPs by leveraging learned priors from large datasets. One class of generative model-based methods involves training models specifically for each IP using paired data (Aggarwal et al., 2018; Mousavi et al., 2015; Yeh et al., 2017). While these approaches can achieve high-quality reconstructions, their generalizability and flexibility are constrained, as they are tailored to specific tasks and often require extensive retraining for new problems. In contrast, another promising direction involves the utilization of pretrained generative models to solve IPs without additional training (Bora et al., 2017; Shah & Hegde, 2018; Liu & Scarlett, 2020b). These methods assume that the underlying signals lie within the range of a variational autoencoder (VAE) (Kingma, 2014) or a generative adversarial network (GAN) (Goodfellow et al., 2014; Karras et al., 2018).

Compared to these conventional generative models like VAEs and GANs, diffusion models (DMs) (Sohl-Dickstein

[1]University of Electronic Science and Technology of China. Correspondence to: Zhaoqiang Liu <zqliu12@gmail.com>.

et al., 2015; Ho et al., 2020; Song & Ermon, 2019; Dhariwal & Nichol, 2021) have recently emerged as a powerful framework for modeling high-dimensional data distributions with remarkable fidelity. These models have demonstrated significant potential in addressing a variety of IPs, often achieving state-of-the-art (SOTA) performance. Motivated by the effectiveness of DMs in solving IPs, this work explores DM based IP solvers, aiming to further advance the field by leveraging the properties and strengths of DMs (Wang et al., 2023; Whang et al., 2022; Saharia et al., 2022b;a; Alkan et al., 2023; Cardoso et al., 2023; Chung et al., 2023a; Feng & Bouman, 2023; Rout et al., 2023; Wu et al., 2023; Aali et al., 2025; Chung et al., 2024; Dou & Song, 2024; Song et al., 2024; Wu et al., 2024; Sun et al., 2024).

## 1.1. Related Work

**Inverse problems with conventional generative models:**
The central idea behind this line of work is to replace sparse priors with generative priors, specifically those from VAEs or GANs. The seminal work (Bora et al., 2017) proposes the CSGM method and demonstrates the use of pre-trained generative priors for solving compressed sensing tasks. The CSGM method aims to minimize $\|\boldsymbol{y} - \mathcal{A}(\boldsymbol{x})\|_2$ over the range of the generative model $\mathcal{G}(\cdot)$, and it has since been extended to various IP through numerous experiments (Oymak et al., 2017; Asim et al., 2020a;b; Liu et al., 2021; Jalal et al., 2021; Liu et al., 2022a;b; Chen et al., 2023b; Liu et al., 2024).

However, there are limitations to these methods. First, the underlying signal $\boldsymbol{x}^*$ often lies outside the range of the generative model, which can result in suboptimal reconstruction performance. Second, these approaches heavily depend on the choice of the initial vector, leading to potential convergence to local minima.

To address these issues, several methods employ projected gradient descent (PGD) (Shah & Hegde, 2018; Hyder et al., 2019; Peng et al., 2020; Liu & Han, 2022), using iterative projections to mitigate the risk of local minima. Some approaches allow for sparse deviations (Dhar et al., 2018) from the range of the generative model to capture signals outside the range. Additionally, the works (Daras et al., 2021; 2022) further perform optimization in intermediate layers to better align the reconstruction with the measurements.

**Inverse problems with diffusion models:** DMs have unlocked new possibilities in a variety of applications due to their ability to generate high-quality samples and model complex data distributions. Since the seminal works (Sohl-Dickstein et al., 2015; Ho et al., 2020; Song & Ermon, 2019), many studies have focused on improving the efficiency and quality of DMs, with efforts aimed at faster sampling speeds and higher output fidelity (Song et al., 2021a; Lu et al.,

2022a;b; Zhao et al., 2024). These frameworks have been widely applied to IPs, achieving remarkable reconstruction performance.

DMs can be specifically trained for particular tasks, such as super-resolution (Gao et al., 2023; Shang et al., 2024), inpainting (Lugmayr et al., 2022), and deblurring (Sanghvi et al., 2025), but their generalization to other tasks remains limited. In this paper, we focus on using pre-trained unconditional DMs and propose a general framework applicable to a range of IPs. The key challenge is how to apply the knowledge of the prior distribution corresponding to the pre-trained DM during the sampling process.

Based on how algorithms reconstruct the underlying signal $\boldsymbol{x}^*$ with prior distribution $p(\boldsymbol{x})$, works with pre-trained DMs can be further classified. Some methods apply Bayes' rule (Fabian et al., 2023; Fei et al., 2023; Zhang et al., 2023) to decompose the conditional score $\nabla_{\boldsymbol{x}} \log p(\boldsymbol{x}|\boldsymbol{y})$ into a tractable unconditional score $\nabla_{\boldsymbol{x}} \log p(\boldsymbol{x})$ provided by the pretrained DMs and an intractable measurement-matching term, performing iterative corrections. For example, DDRM (Kawar et al., 2022) uses singular value decomposition (SVD) to approximate the measurement-matching term in spectral space. MCG (Chung et al., 2022) adapts Tweedie's formula in linear IPs to approximate the reconstructed signal, followed by a gradient-based posterior correction. It also performs projection with a manifold constraint to ensure that corrections remain on the data manifold. DPS (Chung et al., 2023b) discards the projection step in the reverse process of MCG to prevent the samples from falling off the manifold in noisy tasks and generalizes to both linear and nonlinear tasks. ΠGDM (Song et al., 2023) employs pseudo-inverse guidance to encourage data consistency between denoising results and the degraded image. Although these methods achieve high performance in many tasks, they are sometimes sensitive to guidance strength and may fail in simple scenarios, generating unnatural images or incorrect reconstructions.

Another class of methods treats DMs as black-box generative models and focuses on finding the initial latent vector (Wang et al., 2024; Xu et al., 2024). In particular, DM-Plug (Wang et al., 2024) directly optimizes the initial latent vector and demonstrates robust performance across various tasks, with the advantage of not requiring specially designed guidance strengths. However, unlike conventional generative methods that map from the latent to the image space with a single number of function evaluations (NFE), diffusion models require multiple NFEs for the reverse process. This creates a significant memory burden for DMPlug, as larger computational graphs are needed to store information throughout the entire process. Additionally, similar to the CSGM method, if not initialized appropriately, DMPlug may converge to suboptimal points.

## 1.2. Contributions

In this paper, we focus on resolving the memory burden and suboptimal convergence problems prevalent in DM-based CSGM-type methods, such as DMPlug. Given that the multiple sampling steps of DMs correspond to the composition of multiple functions, we initiate by applying Intermediate Layer Optimization (ILO) to relieve the memory burden. Moreover, inspired by the works in (Dhar et al., 2018; Daras et al., 2021), we introduce sparse deviations to broaden the range of DMs. This expansion allows for the exploration of underlying signals beyond the original range of the DM. Additionally, we incorporate the Projected Gradient Descent (PGD) method, iteratively updating the predicted signals to circumvent suboptimal solutions that may arise due to the selection of the initial point. The main contributions of this study are as follows:

- We propose two novel methods, referred to as DMILO and DMILO-PGD. These methods integrate ILO with and without PGD respectively. They effectively (i) reduce the memory burden and (ii) mitigate the suboptimal convergence issues associated with DM-based CSGM-type methods.
- Under appropriate conditions, we offer an intuitive theoretical analysis of the proposed approach. This analysis demonstrates its efficacy in solving IPs.
- Through comprehensive numerical experiments conducted on diverse image datasets, we verify the superior performance of our method compared to SOTA techniques across various linear and nonlinear measurement scenarios.

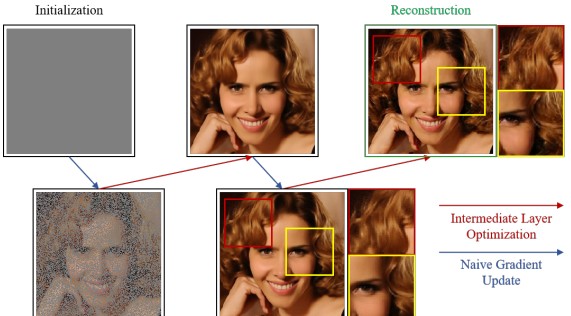

Figure 1: **Illustration of our algorithm**. Starting from a zero vector, we first perform a naive gradient descent update step *(blue arrow)* and then projection step through intermediate layer optimization *(red arrow)* alternatively to reach the final estimate.

## 1.3. Notation

We use upper-case boldface letters to denote matrices and lower-case boldface letters to denote vectors. For $\mathcal{S}_1 \subseteq \mathbb{R}^n$ and $\mathcal{S}_2 \subseteq \mathbb{R}^n$, $\mathcal{S}_1 + \mathcal{S}_2 = \{\boldsymbol{s}_1 + \boldsymbol{s}_2 \ : \ \boldsymbol{s}_1 \in \mathcal{S}_1, \boldsymbol{s}_2 \in \mathcal{S}_2\}$

and $\mathcal{S}_1 - \mathcal{S}_2 = \{\boldsymbol{s}_1 - \boldsymbol{s}_2 \ : \ \boldsymbol{s}_1 \in \mathcal{S}_1, \boldsymbol{s}_2 \in \mathcal{S}_2\}$. We define the $\ell_1$-ball $B_1^n(r) := \{\boldsymbol{x} \in \mathbb{R}^n \ : \ \|\boldsymbol{x}\|_1 \leq r\}$.

## 2. Preliminaries

A diffusion model (DM) comprises a forward process that progressively transforms real data into noise, and a reverse process that aims to invert this transformation and recover the original data distribution. The forward process of DMs can be described by the following the stochastic differential equation (SDE):

$$\mathrm{d}\boldsymbol{x} \ = \ f(t)\,\boldsymbol{x}\,\mathrm{d}t \ + \ g(t)\,\mathrm{d}\boldsymbol{w}_t, \quad \boldsymbol{x}_0 \sim p_0, \qquad (2)$$

where $f(t)$ and $g(t)$ are the drift and diffusion coefficients, respectively, and $\boldsymbol{w}_t$ is a standard Wiener process. For each $t \in [0, T]$, the distribution of $\boldsymbol{x}_t$ conditioned on $\boldsymbol{x}_0$ is Gaussian with $\boldsymbol{x}_t | \boldsymbol{x}_0 \sim \mathcal{N}(\alpha_t \boldsymbol{x}_0, \sigma_t^2\,\mathbf{I})$, where $\alpha_t$ and $\sigma_t$ are positive, differentiable functions with bounded derivatives, and their ratio $\alpha_t^2/\sigma_t^2$ (i.e., the signal-to-noise ratio) is strictly decreasing in $t$. The marginal distribution of $\boldsymbol{x}_t$ is denoted by $p_t$, and $p_{0t}$ denotes the distribution of $\boldsymbol{x}_t$ conditioned on $\boldsymbol{x}_0$. These functions $\alpha_t$ and $\sigma_t$ are selected so that $p_T$ approximates a zero-mean Gaussian with covariance $\tilde{\sigma}^2\mathbf{I}$ for some $\tilde{\sigma} > 0$.

To guarantee that the SDE in (2) indeed induces the transition distribution $p_{0t}$, the drift and diffusion coefficients must satisfy:

$$f(t) \ = \ \frac{\mathrm{d}\log\alpha_t}{\mathrm{d}t}, \quad g^2(t) \ = \ \frac{\mathrm{d}\sigma_t^2}{\mathrm{d}t} - 2\,\frac{\mathrm{d}\log\alpha_t}{\mathrm{d}t}\,\sigma_t^2. \ (3)$$

It is known from (Anderson et al., 1958) that the forward SDE in (2) has a corresponding reverse-time diffusion process evolving from $T$ down to $0$. This reverse SDE can be written as (Ho et al., 2020; Song et al., 2021b):

$$\mathrm{d}\boldsymbol{x}_t \ = \ \big[f(t)\,\boldsymbol{x}_t \ - \ g^2(t)\,\nabla_{\boldsymbol{x}}\log p_t(\boldsymbol{x}_t)\big]\mathrm{d}t \ + \ g(t)\,\mathrm{d}\bar{\boldsymbol{w}}_t, \tag{4}$$

where $\bar{\boldsymbol{w}}_t$ is a Wiener process in reverse time, and $\nabla_{\boldsymbol{x}}\log p_t(\boldsymbol{x}_t)$ is the score function of $p_t$.

As shown by (Song et al., 2021b), one may also construct a deterministic process whose trajectories share the same marginal densities $p_t(\boldsymbol{x})$ as the reverse SDE. This process is governed by the ordinary differential equation (ODE):

$$\mathrm{d}\boldsymbol{x}_t \ = \ \big[f(t)\,\boldsymbol{x}_t \ - \ \tfrac{1}{2}g^2(t)\,\nabla_{\boldsymbol{x}}\log p_t(\boldsymbol{x}_t)\big]\mathrm{d}t. \tag{5}$$

Hence, sampling can be performed via numerical discretization or integration of this ODE.

In practice, the score function $\nabla_{\boldsymbol{x}}\log p_t(\boldsymbol{x}_t)$ is approximated by a neural network. A common approach is to train a noise prediction network $\boldsymbol{\epsilon}_\theta(\boldsymbol{x}_t, t)$ to estimate the scaled

score $-\sigma_t \nabla_{\boldsymbol{x}} \log p_t(\boldsymbol{x}_t)$. The parameter set $\theta$ is optimized by minimizing the following objective:

$$\int_0^T \mathbb{E}_{\boldsymbol{x}_0 \sim p_0} \mathbb{E}_{\boldsymbol{\epsilon} \sim \mathcal{N}(\mathbf{0},\mathbf{I})} \left[ \|\boldsymbol{\epsilon}_\theta(\alpha_t \boldsymbol{x}_0 + \sigma_t \boldsymbol{\epsilon}, t) - \boldsymbol{\epsilon}\|_2^2 \right] \mathrm{d}t. \tag{6}$$

By substituting $\nabla_{\boldsymbol{x}} \log p_t(\boldsymbol{x}_t) = -\boldsymbol{\epsilon}_\theta(\boldsymbol{x}_t, t)/\sigma_t$ into (5) and exploiting its semi-linear structure, one obtains the numerical integration formula (Lu et al., 2022a):

$$\boldsymbol{x}_t = e^{\int_s^t f(\tau)\,\mathrm{d}\tau} \boldsymbol{x}_s + \int_s^t e^{\int_\tau^t f(r)\mathrm{d}r} \frac{g^2(\tau)}{2\,\sigma_\tau} \boldsymbol{\epsilon}_\theta(\boldsymbol{x}_\tau, \tau)\,\mathrm{d}\tau. \tag{7}$$

An alternative strategy is to train a data prediction network $\boldsymbol{x}_\theta(\boldsymbol{x}_t, t)$ such that $\boldsymbol{\epsilon}_\theta(\boldsymbol{x}_t, t) = (\boldsymbol{x}_t - \alpha_t \boldsymbol{x}_\theta(\boldsymbol{x}_t, t))/\sigma_t$. Then, by substituting (3) into (7) and using the relation between $\boldsymbol{\epsilon}_\theta$ and $\boldsymbol{x}_\theta$, one obtains the following integration formula (Lu et al., 2022b):

$$\boldsymbol{x}_t = \frac{\sigma_t}{\sigma_s} \boldsymbol{x}_s + \sigma_t \int_{\lambda_s}^{\lambda_t} \exp(\lambda)\, \hat{\boldsymbol{x}}_\theta(\hat{\boldsymbol{x}}_\lambda, \lambda)\,\mathrm{d}\lambda, \tag{8}$$

where $\lambda_t := \log(\alpha_t/\sigma_t)$ is strictly decreasing in $t$ and admits an inverse function $t_\lambda(\cdot)$, and thus the data prediction network $\boldsymbol{x}_\theta(\boldsymbol{x}_{t_\lambda(\lambda)}, t_\lambda(\lambda))$ can be rewritten as $\hat{\boldsymbol{x}}_\theta(\hat{\boldsymbol{x}}_\lambda, \lambda)$.

In the sampling process, DMs iteratively denoise an initial Gaussian noise vector to produce an image sample. Concretely, let $[\epsilon, T]$ be partitioned by $\epsilon = t_0 < t_1 < \cdots < t_{N-1} < t_N = T$, where $\epsilon > 0$ is small to avoid numerical instabilities (Lu et al., 2022a). Then, by applying a first-order approximation to (8), the transition from $t_i$ to $t_{i-1}$ can be written as:

$$\tilde{\boldsymbol{x}}_{t_{i-1}} = \frac{\sigma_{t_{i-1}}}{\sigma_{t_i}} \tilde{\boldsymbol{x}}_{t_i} + \sigma_{t_{i-1}} \left( \frac{\alpha_{t_{i-1}}}{\sigma_{t_{i-1}}} - \frac{\alpha_{t_i}}{\sigma_{t_i}} \right) \boldsymbol{x}_\theta(\tilde{\boldsymbol{x}}_{t_i}, t_i). \tag{9}$$

This update rule matches the widely used DDIM sampling method (Song et al., 2021a). Note that the $i$-th sampling step corresponds to the following function:

$$g_i(\boldsymbol{x}) := \frac{\sigma_{t_{i-1}}}{\sigma_{t_i}} \boldsymbol{x} + \sigma_{t_{i-1}} \left( \frac{\alpha_{t_{i-1}}}{\sigma_{t_{i-1}}} - \frac{\alpha_{t_i}}{\sigma_{t_i}} \right) \boldsymbol{x}_\theta(\boldsymbol{x}, t_i), \tag{10}$$

and the entire sampling process corresponds to the composition of $g_i$ for $i = N, N-1, \ldots, 1$.

## 3. Methods

In this section, we introduce our proposed methods, DMILO and DMILO-PGD, two novel approaches for solving IPs using pretrained DMs. In Section 3.1, we present the concept of ILO and show how it can be naturally integrated into DMs to address the high memory usage found in recent DM-based CSGM methods such as DMPlug. In Section 3.2,

we adapt the PGD framework to our setting, thereby alleviating potential suboptimal solutions. In doing so, we also highlight an illustrative example that explains why using the forward model during projection yields more robust results compared to existing approaches.

### 3.1. DMILO

ILO was originally introduced to improve the ability of conventional deep generative models to conform to given measurements (Daras et al., 2021). The key idea is to split the generative model into intermediate layers and iteratively optimize the latent representations at each layer. Specifically, for a conventional generative model such as GAN, even if it only requires a single sampling step to map noise to image, ILO decomposes it into four intermediate layers and optimizes the representation at each intermediate layer. However, this decomposition is often architecture-dependent and empirically determined, making it hard to generalize across diverse models.

In contrast, when viewing the entire sampling process of a DM as a generative function $\mathcal{G}(\cdot)$, it is naturally decomposed into a composition of simpler functions:

$$\mathcal{G}(\cdot) = g_1 \circ g_2 \circ \cdots \circ g_N(\cdot), \tag{11}$$

where $N$ denotes the total number of sampling steps, and $g_i(\cdot)$ represents the function corresponding to the $i$-th sampling step (see, e.g., (10)). Such a composition is independent of the architecture of the denoising network or sampling scheme, making it straightforward to be plugged into any DM. In this paper, we follow the DDIM setting and use Eq. (10) as the sampling function.

Based on this composition, our method operates iteratively. In each iteration, we first optimize the input $\hat{\boldsymbol{x}}_{t_1}$ to the final function $g_1(\cdot)$. Here, we use a technique inspired by (Dhar et al., 2018), and introduce a sparse vector $\hat{\boldsymbol{\nu}}_{t_1}$ to search for the underlying signal outside the range of the generator:

$$\hat{\boldsymbol{x}}_{t_1}, \hat{\boldsymbol{\nu}}_{t_1} = \arg\min_{\boldsymbol{x},\boldsymbol{\nu}} \|\mathcal{A}(\hat{\boldsymbol{x}}_{t_0}) - \mathcal{A}(g_1(\boldsymbol{x}) + \boldsymbol{\nu})\|_2^2 + \lambda \|\boldsymbol{\nu}\|_1, \tag{12}$$

where $\lambda$ is the Lagrange multiplier, $\mathcal{A}(\cdot)$ denotes the forward operator, and $\hat{\boldsymbol{x}}_{t_0}$ is the estimated signal. Since the estimated signal is unavailable and often noisy, we replace $\mathcal{A}(\hat{\boldsymbol{x}}_{t_0})$ with the observed vector $\boldsymbol{y}$ in practice. As the minimization problem in Eq.(12) is highly non-convex and obtaining its globally optimal solutions is not feasible, we approximately solve this optimization problem using the Adam optimizer and apply $\ell_2$-regularization to $\hat{\boldsymbol{x}}_{t_1}$ to avoid overfitting.

We then proceed iteratively for the remaining layers (or sampling steps). Specifically, for each subsequent layer, we perform the optimization as follows:

$$\hat{\boldsymbol{x}}_{t_i}, \hat{\boldsymbol{\nu}}_{t_i} = \arg\min_{\boldsymbol{x},\boldsymbol{\nu}} \|\hat{\boldsymbol{x}}_{t_{i-1}} - (g_i(\boldsymbol{x}) + \boldsymbol{\nu})\|_2^2 + \lambda \|\boldsymbol{\nu}\|_1. \tag{13}$$

Unlike other DM-based CSGM-type approaches that must retain the entire gradient graph throughout the sampling process, our method requires only the gradient information of a single sampling step at a time. This offers a substantial *reduction in memory usage* and can be seamlessly applied to different sampling strategies. Additionally, incorporating the sparse deviation term extends the range of the generator and can yield better reconstruction quality in practice. For convenience, the iterative procedure for DMILO is presented in Algorithm 1.

---

**Algorithm 1** DMILO

---

1: **Input:** Diffusion denoising model $\mathcal{G}(\cdot) = g_1 \circ \cdots \circ g_{N-1} \circ g_N(\cdot)$, total number of sampling steps $N$, noisy observation $\boldsymbol{y}$, forward model $\mathcal{A}(\cdot)$, iteration steps $J$, Lagrange multiplier $\lambda$
2: Initialize $\boldsymbol{x}_{t_N}^{(0)} \sim \mathcal{N}(\boldsymbol{0}, \boldsymbol{I})$ and $\boldsymbol{\nu}_{t_N}^{(0)} = \boldsymbol{0}$
3: **for** $i = N$ **to** $2$ **do**
4:   $\boldsymbol{x}_{t_{i-1}}^{(0)} = g_i(\boldsymbol{x}_{t_i}), \boldsymbol{\nu}_{t_{i-1}}^{(0)} = \boldsymbol{0}$
5: **end for**
6: **for** $j = 1$ **to** $J$ **do**
7:   Approximately solve $(\boldsymbol{x}_{t_1}^{(j)}, \boldsymbol{\nu}_{t_1}^{(j)}) = \arg\min_{(\boldsymbol{x}, \boldsymbol{\nu})} \|\boldsymbol{y} - \mathcal{A}(g_1(\boldsymbol{x}) + \boldsymbol{\nu})\|_2^2 + \lambda \|\boldsymbol{\nu}\|_1$ using the Adam optimizer, initialized at $(\boldsymbol{x}_{t_1}^{(j-1)}, \boldsymbol{\nu}_{t_1}^{(j-1)})$
8:   **for** $i = 2$ **to** $N$ **do**
9:     Approximately solve $(\boldsymbol{x}_{t_i}^{(j)}, \boldsymbol{\nu}_{t_i}^{(j)}) = \arg\min_{(\boldsymbol{x}, \boldsymbol{\nu})} \|\boldsymbol{x}_{t_{i-1}}^{(j)} - (g_i(\boldsymbol{x}) + \boldsymbol{\nu})\|_2^2 + \lambda \|\boldsymbol{\nu}\|_1$ using the Adam optimizer, initialized at $(\boldsymbol{x}_{t_i}^{(j-1)}, \boldsymbol{\nu}_{t_i}^{(j-1)})$
10:   **end for**
11:   **for** $i = N$ **to** $1$ **do**
12:     $\boldsymbol{x}_{t_{i-1}}^{(j)} = g_i(\boldsymbol{x}_{t_i}^{(j)}) + \boldsymbol{\nu}_{t_i}^{(j)}$
13:   **end for**
14: **end for**
15: **Return** $\hat{\boldsymbol{x}} = \boldsymbol{x}_{t_0}^{(J)}$

---

We also compare the memory usage of our methods with DMPlug using a 2.75 GB diffusion model on an NVIDIA RTX 4090. As shown in Table 1, our approaches significantly reduce memory consumption and maintain a consistent memory footprint as the number of sampling steps increase, underscoring both their practical efficiency in real-world settings and their extensibility to different sampling methods and diffusion models.

### 3.2. DMILO-PGD

We now describe how to adapt PGD to further address suboptimal solutions. PGD alternates between a gradient descent step and a projection onto the range of the generator (Shah & Hegde, 2018). In our approach, we replace the projection step with DMILO, as outlined in Algorithm 2. Difference from conventional PGD which sorely minimize

Table 1: Memory usage (in GB) for DMPlug, DMILO, and DMILO-PGD across different sampling steps. The model size is 2.75 GB for the super-resolution task on the LSUN Bedroom dataset (Yu et al., 2015). "N/A" indicates that the method exceeds available memory.

| Sampling Steps | DMPlug | DMILO | DMILO-PGD |
|---|---|---|---|
| 1 | 10.53 | 10.53 | 10.53 |
| 2 | 15.72 | 10.53 | 10.54 |
| 3 | 20.83 | 10.53 | 10.54 |
| 4 | N/A | 10.54 | 10.54 |

$\|\mathcal{G}(\boldsymbol{x}_{t_N}) - \hat{\boldsymbol{x}}_{t_0}\|_2^2$ without participant of forward operator $\mathcal{A}(\cdot)$, we minimize $\|\mathcal{A}(\mathcal{G}(\boldsymbol{x}_{t_N})) - \mathcal{A}(\hat{\boldsymbol{x}}_{t_0})\|_2^2$, where $\hat{\boldsymbol{x}}_{t_0}$ denotes the result from the gradient descent update (see Step 5 in Algorithm 2). The efficacy of leveraging the forward operator to guide the projection has been illustrated in Figure 2, and the corresponding theoretical evidence has been provided in Theorem 4.4.

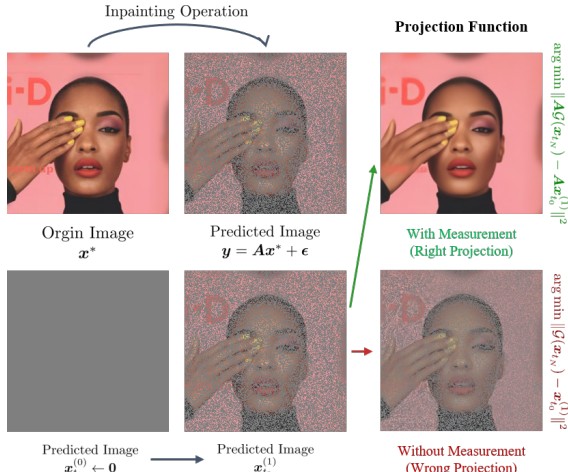

Figure 2: **Illustration of two projection strategies in DMILO-PGD.** The *top-right* is the reconstructed image using our method that leverages the forward operator to guide the projection (**green arrow**). In contrast, the *bottom-right* demonstrates how a purely "distance-based" projection (**red arrow**) can yield a problematic reconstruction.

## 4. Theoretical Analysis

In this section, we present an intuitive theoretical analysis of the effectiveness of the proposed DMILO-PGD method. As observed from the experimental results, this method performs reasonably well for various instances of IPs. For the sake of simplicity, we build upon the results in (Daras et al., 2021) and consider only the case where the generator $\mathcal{G} : \mathbb{R}^n \to \mathbb{R}^n$ is the composition of two functions with

**Algorithm 2** DMILO-PGD

1: **Input:** Diffusion denoising model $\mathcal{G}(\cdot) = g_1 \circ \cdots \circ g_{N-1} \circ g_N(\cdot)$, total number of sampling steps $N$, noisy observation $\boldsymbol{y}$, forward model $\mathcal{A}(\cdot)$, iteration steps $E$, learning rate $\eta$, Lagrange multiplier $\lambda$
2: Initialize $\boldsymbol{x}_{t_0}^{(0)} = \boldsymbol{0}$, $\boldsymbol{x}_{t_N}^{(0)} \sim \mathcal{N}(\boldsymbol{0}, \boldsymbol{I})$ and $\boldsymbol{\nu}_{t_N}^{(0)} = \boldsymbol{0}$
3: **for** $i = N$ **to** $2$ **do**
4: $\quad \boldsymbol{x}_{t_{i-1}}^{(0)} = g_i(\boldsymbol{x}_{t_i}), \boldsymbol{\nu}_{t_{i-1}}^{(0)} = \boldsymbol{0}$
5: **end for**
6: **for** $e = 1$ **to** $E$ **do**
7: $\quad \boldsymbol{x}_{t_0}^{(e)} = \boldsymbol{x}_{t_0}^{(e-1)} - \eta\nabla\|\boldsymbol{y} - \mathcal{A}(\boldsymbol{x}_{t_0}^{(e-1)})\|_2^2$
8: $\quad$ Approximately solve $(\boldsymbol{x}_{t_1}^{(e)}, \boldsymbol{\nu}_{t_1}^{(e)}) = \arg\min_{(\boldsymbol{x},\boldsymbol{\nu})} \|\mathcal{A}(\boldsymbol{x}_{t_0}^{(e)}) - \mathcal{A}(g_1(\boldsymbol{x}) + \boldsymbol{\nu})\|_2^2 + \lambda\|\boldsymbol{\nu}\|_1$ using the Adam optimizer, initialized at $(\boldsymbol{x}_{t_1}^{(e-1)}, \boldsymbol{\nu}_{t_1}^{(e-1)})$
9: $\quad$ **for** $i = 2$ **to** $N$ **do**
10: $\quad\quad$ Approximately solve $(\boldsymbol{x}_{t_i}^{(e)}, \boldsymbol{\nu}_{t_i}^{(e)}) = \arg\min_{(\boldsymbol{x},\boldsymbol{\nu})} \|\boldsymbol{x}_{t_{i-1}}^{(e)} - (g_i(\boldsymbol{x}) + \boldsymbol{\nu})\|_2^2 + \lambda\|\boldsymbol{\nu}\|_1$ using the Adam optimizer, initialized at $(\boldsymbol{x}_{t_i}^{(e-1)}, \boldsymbol{\nu}_{t_i}^{(e-1)})$
11: $\quad$ **end for**
12: $\quad$ **for** $i = N$ **to** $1$ **do**
13: $\quad\quad \boldsymbol{x}_{t_{i-1}}^{(e)} = g_i(\boldsymbol{x}_{t_i}^{(e)}) + \boldsymbol{\nu}_{t_i}^{(e)}$
14: $\quad$ **end for**
15: **end for**
16: **Return** $\hat{\boldsymbol{x}} = \boldsymbol{x}_{t_0}^{(E)}$

---

$\mathcal{G} = g_1 \circ g_2$, and we restrict our attention to the linear task, where the measurement model in (1) simplifies to:

$$\boldsymbol{y} = \boldsymbol{A}\boldsymbol{x}^* + \boldsymbol{\epsilon}. \tag{14}$$

Here, $\boldsymbol{A} \in \mathbb{R}^{m \times n}$ represents the linear measurement matrix.

In theoretical studies of DMs, it is standard to assume that the denoising network $\boldsymbol{x}_{\boldsymbol{\theta}}(\boldsymbol{x}, t)$ is Lipschitz continuous with respect to its first argument (Chen et al., 2022; 2023a; Li et al., 2024). Then, given that $g_1$ corresponds to a sampling step in DMs, it is reasonable to make the following assumption:

**Assumption 4.1.** The function $g_1$ is $L_1$-Lipschitz continuous for some $L_1 > 0$.

Furthermore, following the setting in (Li & Yan, 2024), and inspired by the common characteristic of natural image distributions that the underlying target distribution is concentrated on or near low-dimensional manifolds within the high-dimensional ambient space, we make the following low-dimensionality assumption regarding the range of $g_2$.

**Assumption 4.2.** Let $\mathcal{X}_2$ be the range of $g_2$, i.e., $\mathcal{X}_2 = g_2(\mathbb{R}^n)$. For any $\delta > 0$, we define the intrinsic dimension of $\mathcal{X}_2$ as a positive quantity $k_2$ (typically a positive integer

much smaller than $n$) such that[1]

$$\log N_\delta(\mathcal{X}_2) \leq C_{g_2} k_2 \log\left(\frac{n}{\delta}\right), \tag{15}$$

where $C_{g_2}$ is a positive constant that depends on $g_2$.

Next, we introduce the Set-Restricted Eigenvalue Condition (S-REC), which has been extensively explored in CSGM-type methods (Bora et al., 2017).

**Definition 4.3.** Let $\mathcal{S} \subseteq \mathbb{R}^n$. For some parameters $\gamma \in (0, 1), \delta \geq 0$, a matrix $\boldsymbol{A} \in \mathbb{R}^{m \times n}$ is said to satisfy the S-REC$(\mathcal{S}, \gamma, \delta)$ if for all $\boldsymbol{s}_1, \boldsymbol{s}_2 \in \mathcal{S}$,

$$\|\boldsymbol{A}(\boldsymbol{s}_1 - \boldsymbol{s}_2)\|_2 \geq \gamma\|\boldsymbol{s}_1 - \boldsymbol{s}_2\|_2 - \delta. \tag{16}$$

By proving that the Gaussian measurement matrix satisfies the S-REC over the set $g_1(\mathcal{X}_2 + B_1^n(r))$ for some $r > 0$, we adapt the result from (Daras et al., 2021, Theorem 1) to derive the following theorem.

**Theorem 4.4.** *Suppose that Assumptions 4.1 and 4.2 hold. Let $\boldsymbol{A} \in \mathbb{R}^{m \times n}$ have i.i.d. $\mathcal{N}(0, 1/m)$ entries.[2] For fixed $\gamma \in (0, 1)$, $\delta > 0$, and $k \in (0, \sqrt{n})$, let $r = \frac{k\delta}{L_1}$. Consider the true optimum within the extended range:*

$$\bar{\boldsymbol{x}}_1 := \arg\min_{\boldsymbol{x}_1 \in \mathcal{X}_2 + B_1^n(r)} \|\boldsymbol{x}^* - g_1(\boldsymbol{x}_1)\|_2, \tag{17}$$

*and the measurement optimum within the extended range:[3]*

$$\hat{\boldsymbol{x}}_1 := \arg\min_{\boldsymbol{x}_1 \in \mathcal{X}_2 + B_1^n(r)} \|\boldsymbol{A}\boldsymbol{x}^* - \boldsymbol{A}g_1(\boldsymbol{x}_1)\|_2. \tag{18}$$

*Then, if the number of measurements $m$ is sufficiently large with $m = \Omega(k_2 \log\frac{L_1 n}{\delta} + k^2 \log(3n))$, we have with probability $1 - e^{-\Omega(m)}$ that*

$$\|g_1(\hat{\boldsymbol{x}}_1) - \boldsymbol{x}^*\|_2 \leq \left(1 + \frac{3}{\gamma}\right) \cdot \|g_1(\bar{\boldsymbol{x}}_1) - \boldsymbol{x}^*\|_2 + \frac{\delta}{\gamma}. \tag{19}$$

Similar to Theorem 1 in (Daras et al., 2021), our Theorem 4.4 essentially states that if the number of measurements $m$ is sufficiently large, the measurement optimum $\hat{\boldsymbol{x}}_1$ is nearly as good as the true optimum $\bar{\boldsymbol{x}}_1$. It is important to note

---

[1]For $\mathcal{X} \in \mathbb{R}^n$ and $\epsilon > 0$, a subset $\mathcal{S} \subseteq \mathcal{X}$ is said be an $\epsilon$-net of $\mathcal{X}$ if, for all $\boldsymbol{x} \in \mathcal{X}$, there exists some $\boldsymbol{s} \in \mathcal{S}$ such that $\|\boldsymbol{s} - \boldsymbol{x}\|_2 \leq \epsilon$. The minimal cardinality of an $\epsilon$-net of $\mathcal{X}$ (assuming it is finite) is denoted by $N_\epsilon(\mathcal{X})$ and is called the covering number of $\mathcal{X}$ (with parameter $\epsilon$).

[2]As noted in (Daras et al., 2021), analogous results hold when $\boldsymbol{A}$ is a partial circulant matrix, a case that is highly relevant to the blind image deblurring problem. In the context of partial circulant matrices, the number of measurements $m$ needed to establish the S-REC is of nearly the same order (neglecting additional logarithmic factors).

[3]Note that the optimization problem in (18) can be regarded as a variant of (12), and the optimization problem in (17) can be regarded as a variant of (13).

that a reconstruction algorithm can only access the measurement error and can never compute $\bar{x}_1$. Therefore, Theorem 4.4 also provides theoretical support for performing (12) instead of (13) in DMILO-PGD. The proof of Theorem 4.4 is deferred to Appendix A.

# 5. Experiments

In this section, we mainly adhere to the settings in DM-Plug (Wang et al., 2024) to assess our DMILO and DMILO-PGD methods. We compare our method with several recent baseline approaches on four linear inverse problems (IP) tasks, namely super-resolution, inpainting, and two linear image deblurring tasks involving Gaussian deblurring and motion deblurring. Furthermore, we evaluate on two nonlinear IP tasks, including nonlinear deblurring and blind image deblurring (BID). To measure recovery quality, we follow (Blau & Michaeli, 2018) and use three standard metrics: Peak Signal-to-Noise Ratio (PSNR) and Structural Similarity Index (SSIM), which evaluate distortion, and Learned Perceptual Image Patch Similarity (LPIPS) (Zhang et al., 2018) with the default backbone, which measures perceptual quality. Additionally, we compute the Fréchet Inception Distance (FID) on the linear image deblurring task to better illustrate the perceptual quality of the generated images. We use **bold** to indicate the best performance, underline for the second-best, green to denote performance improvement, and red to signify performance decline.

## 5.1. Inpainting and Super-resolution

For inpainting and super-resolution two linear tasks, we create evaluation sets by sampling 100 images each from CelebA (Liu et al., 2015), FFHQ (Karras et al., 2019), and LSUN-bedroom (Yu et al., 2015).[4] All images are resized to $256 \times 256 \times 3$ pixels. For super-resolution, we generate measurements by applying $4\times$ bicubic downsampling, and for inpainting, we use a random mask with 70% missing pixels. Consistent with the setting in DMPlug (Wang et al., 2024), all measurements are corrupted by additive zero - mean Gaussian noise with a standard deviation of $\sigma = 0.01$. We compare our methods with DDRM (Kawar et al., 2022), DPS (Chung et al., 2023b), ΠGDM (Song et al., 2023), RED-diff (Song et al., 2023), DAPS (Zhang et al., 2025), DiffPIR (Zhu et al., 2023) and DMPlug (Wang et al., 2024).

In the inpainting and super-resolution tasks, both DMILO and DMILO-PGD are configured with a Lagrange multiplier $\lambda = 0.1$. For inpainting, during the optimization process, both DMILO and DMILO-PGD employ the Adam optimizer with an inner learning rate of 0.02. They run for 200 inner iterations and 5 outer iterations. In the case of DMILO-PGD,

---

[4]The experimental results for LSUN-bedroom are presented in Appendix C due to the page limit.

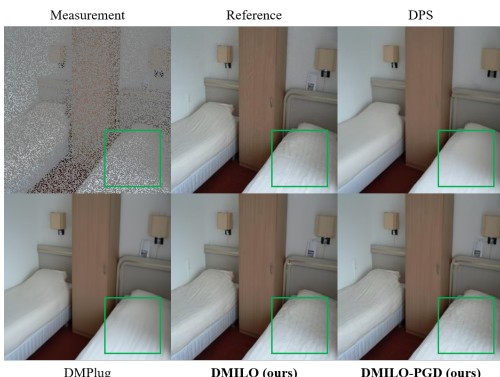

Figure 3: Visualization of sample results from our methods for inpainting with additive Gaussian noise ($\sigma = 0.01$).

the outer learning rate $\eta$ is set to 0.5. For super-resolution, an inner learning rate of 0.02 is also used, along with 400 inner iterations and 10 outer iterations. For DMILO-PGD in super-resolution, the outer learning rate $\eta$ is set to 8. The quantitative results are presented in Table 2. Evidently, our DMILO-PGD approach surpasses all competing methods in most metrics. This clearly demonstrates its superior reconstruction performance, highlighting its effectiveness and competitiveness in these two linear IP tasks.

## 5.2. Linear Image Deblurring

For linear image deblurring, we choose Gaussian deblurring and motion deblurring two linear image deblurring tasks. To validate the performance of different methods, we randomly sample 100 images from CelebA, FFHQ, and ImageNet (Deng et al., 2009). All images are resized to $256 \times 256 \times 3$ and the noise level $\sigma$ is set to 0.01 similarly. We compare our methods with DPS, RED-diff, DPIR (Zhang et al., 2021) and DiffPIR, DMPlug. Note that we follow DMPlug to set the kernel size to $64 \times 64$, which differs from the kernel size initially employed in DPIR and DiffPIR. Such a difference might impact the results. The results presented in Tables 3 and 4 indicates that our methods generally perform well in terms of both realism and reconstruction metrics, demonstrating its ability to effectively balance perceptual quality and distortion. They achieve top performance across all metrics for motion deblurring and achieve either the best or competitive results on Gaussian deblurring.

## 5.3. Nonlinear Deblurring

For nonlinear deblurring, we use the learned blurring operators from (Tran et al., 2021) with a known Gaussian-shaped kernel and additive zero-mean Gaussian noise with a standard deviation of $\sigma = 0.01$, following the setup of DMPlug. Similar to the process for linear tasks, our evaluation sets are created by sampling 100 images from CelebA, FFHQ,

Table 2: (**Linear IPs**) **Super-resolution** and **inpainting** with additive Gaussian noise ($\sigma = 0.01$).

| | Super-resolution ($4\times$) | | | | | | Inpainting (Random $70\%$) | | | | | |
| | CelebA ($256 \times 256$) | | | FFHQ ($256 \times 256$) | | | CelebA ($256 \times 256$) | | | FFHQ ($256 \times 256$) | | |
| | LPIPS↓ | PSNR↑ | SSIM↑ | LPIPS↓ | PSNR↑ | SSIM↑ | LPIPS↓ | PSNR↑ | SSIM↑ | LPIPS↓ | PSNR↑ | SSIM↑ |
|---|---|---|---|---|---|---|---|---|---|---|---|---|
| DDRM | 0.103 | 31.84 | 0.879 | 0.127 | 30.58 | 0.862 | 0.142 | 28.24 | 0.822 | 0.151 | 27.19 | 0.808 |
| DPS | 0.143 | 27.15 | 0.753 | 0.163 | 25.93 | 0.720 | 0.073 | 33.07 | 0.897 | 0.089 | 31.59 | 0.879 |
| ΠGDM | 0.112 | 32.45 | 0.888 | **0.079** | 30.96 | **0.876** | 0.104 | 32.29 | 0.882 | 0.080 | 31.10 | 0.864 |
| RED-diff | 0.105 | 32.48 | 0.888 | 0.127 | 31.09 | 0.865 | 0.238 | 28.19 | 0.800 | 0.224 | 27.38 | 0.789 |
| DAPS | 0.111 | 29.98 | 0.814 | 0.149 | 28.67 | 0.803 | 0.051 | 32.40 | 0.886 | 0.046 | 30.95 | 0.879 |
| DiffPIR | 0.107 | 29.48 | 0.796 | 0.121 | 27.95 | 0.778 | 0.098 | 31.20 | 0.867 | 0.100 | 29.59 | 0.854 |
| DMPlug | 0.127 | 32.38 | 0.875 | 0.124 | 31.22 | 0.866 | 0.066 | 35.51 | 0.935 | 0.068 | 34.16 | 0.927 |
| **DMILO** | 0.133 | 30.81 | 0.785 | 0.129 | 30.26 | 0.789 | 0.025 | 36.07 | 0.951 | 0.029 | 34.23 | 0.937 |
| **DMILO-PGD** | **0.056** | **33.58** | **0.906** | 0.117 | **31.38** | 0.868 | **0.023** | **36.42** | **0.952** | **0.026** | **34.62** | **0.940** |
| **Ours vs. Best comp.** | −0.047 | +1.10 | +0.018 | +0.038 | +0.06 | −0.008 | −0.028 | +0.91 | +0.017 | −0.020 | +0.46 | +0.013 |

Table 3: (**Linear IPs**) **Gaussian deblurring** with additive Gaussian noise ($\sigma = 0.01$).

| | CelebA | | | | FFHQ | | | | ImageNet | | | |
| | FID↓ | LPIPS↓ | PSNR↑ | SSIM↑ | FID↓ | LPIPS↓ | PSNR↑ | SSIM↑ | FID↓ | LPIPS↓ | PSNR↑ | SSIM↑ |
|---|---|---|---|---|---|---|---|---|---|---|---|---|
| DPS | 62.82 | 0.109 | 27.65 | 0.752 | 91.45 | 0.150 | 25.56 | 0.717 | 147.99 | 0.338 | 23.30 | 0.595 |
| RED-diff | 85.14 | 0.221 | 29.59 | 0.808 | 111.52 | 0.272 | 27.15 | 0.778 | 166.04 | 0.497 | 25.07 | 0.666 |
| DPIR | 92.05 | 0.256 | **31.30** | **0.861** | 134.07 | 0.271 | 29.06 | 0.844 | 151.86 | 0.415 | **26.67** | **0.741** |
| DiffPIR | **50.06** | **0.092** | 28.91 | 0.791 | 86.40 | 0.119 | 26.88 | 0.769 | 145.58 | 0.422 | 24.59 | 0.581 |
| DMPlug | 77.06 | 0.172 | 29.70 | 0.776 | 95.46 | 0.181 | 28.27 | 0.806 | **128.74** | 0.324 | 25.20 | 0.662 |
| **DMILO** | 54.42 | **0.092** | 30.89 | 0.816 | **72.34** | **0.110** | **29.60** | **0.852** | 142.13 | **0.316** | 24.81 | 0.702 |
| **DMILO-PGD** | 80.69 | 0.157 | 30.74 | 0.811 | 111.17 | 0.176 | 28.65 | 0.799 | 164.42 | 0.411 | 26.03 | 0.706 |
| **Ours vs. Best comp.** | +4.36 | -0.000 | -0.41 | -0.045 | -14.06 | -0.009 | +0.54 | 0.008 | -13.39 | -0.008 | -0.64 | -0.035 |

and LSUN. Each image is resized to $256 \times 256 \times 3$ pixels. As DDRM is only designed for linear IPs, we compare our DMILO and DMILO-PGD methods against DPS, ΠGDM, RED-diff, and DMPlug.

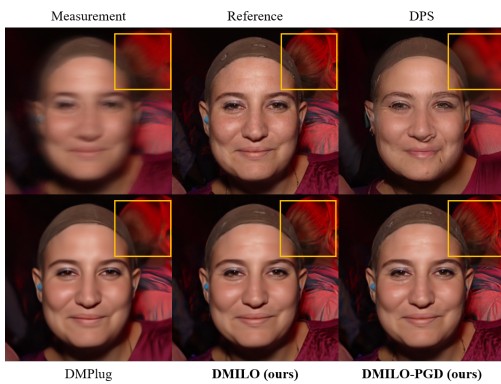

Figure 4: Visualization of sample results for nonlinear deblurring with additive Gaussian noise ($\sigma = 0.01$).

For the nonlinear deblurring task, both DMILO and DMILO-PGD use $\lambda = 0.1$, the Adam optimizer with an inner learning rate of 0.02, 200 inner iterations, and 5 outer iterations. DMILO-PGD further adopts an outer learning rate $\eta = 0.3$. As shown in Table 5, both methods outperform all competitors across all metrics on both datasets, demonstrating the superiority of DMILO and DMILO-PGD for nonlinear deblurring.

### 5.4. Blind Image Deblurring

The BID problem aims to recover an underlying image $x^*$ and an unknown blur kernel $k^*$ from noisy observations $y = k^* * x^* + \epsilon$ where "$*$" denotes convolution, $k^*$ is a spatially invariant kernel, and $\epsilon$ is the noise vector. In our experiments, we consider two types of blur kernels, namely motion kernels and Gaussian kernels, which are the same types used in the linear image deblurring task. To form the evaluation sets, we sample 100 images each from CelebA and FFHQ. We compare our DMILO and DMILO-PGD methods against three recently proposed methods for blind IPs: ILVR (Choi et al., 2021), Blind-DPS (Chung et al., 2023a), and DMPlug.

For both DMILO and DMILO-PGD, the Lagrange multiplier $\lambda$ is set to 0.1. In BID tasks, DMILO employs the Adam optimizer with an inner learning rate of 0.01, 200 inner iterations, and 10 outer iterations. DMILO-PGD uses Adam with an inner learning rate of 0.02, an outer learning rate $\eta = 2$, 500 inner iterations, and 5 outer iterations. The experimental results are presented in Tables 6 and 7. These results clearly demonstrate that DMILO outperforms other methods across all tasks and metrics, underscoring its superior performance in the BID task. However, DMILO-PGD

Table 4: (**Linear IPs**) **Motion deblurring** with additive Gaussian noise ($\sigma = 0.01$).

| | CelebA | | | | FFHQ | | | | ImageNet | | | |
|---|---|---|---|---|---|---|---|---|---|---|---|---|
| | FID↓ | LPIPS↓ | PSNR↑ | SSIM↑ | FID↓ | LPIPS↓ | PSNR↑ | SSIM↑ | FID↓ | LPIPS↓ | PSNR↑ | SSIM↑ |
| DPS | 67.21 | 0.126 | 26.62 | 0.730 | 103.39 | 0.167 | 24.34 | 0.676 | 167.71 | 0.327 | 22.90 | 0.590 |
| RED-diff | 111.34 | 0.229 | 27.32 | 0.758 | 130.81 | 0.272 | 25.40 | 0.730 | 247.22 | 0.494 | 22.51 | 0.587 |
| DPIR | 120.62 | 0.192 | 31.09 | 0.826 | 116.72 | 0.181 | 29.67 | 0.820 | 134.73 | 0.289 | 26.98 | 0.758 |
| DiffPIR | 78.03 | 0.117 | 28.35 | 0.773 | 97.69 | 0.137 | 26.41 | 0.740 | 115.74 | 0.282 | 24.79 | 0.608 |
| DMPlug | 78.57 | 0.164 | 30.25 | 0.824 | 93.66 | 0.173 | 28.58 | 0.812 | 99.87 | 0.285 | 25.49 | 0.696 |
| **DMILO** | **31.08** | **0.044** | **34.15** | **0.908** | **41.48** | **0.044** | **33.21** | **0.909** | **53.77** | **0.098** | **29.67** | **0.841** |
| **DMILO-PGD** | 36.75 | 0.067 | 33.41 | 0.884 | 49.57 | 0.079 | 31.66 | 0.857 | 85.51 | 0.183 | 27.60 | 0.755 |
| **Ours vs. Best comp.** | -36.13 | -0.073 | +3.06 | +0.082 | -52.18 | -0.093 | +3.54 | +0.089 | -46.10 | -0.184 | +2.69 | +0.083 |

Table 5: (**Nonlinear IP**) **Nonlinear deblurring** with additive Gaussian noise ($\sigma = 0.01$).

| | CelebA ($256 \times 256$) | | | FFHQ ($256 \times 256$) | | |
|---|---|---|---|---|---|---|
| | LPIPS↓ | PSNR↑ | SSIM↑ | LPIPS↓ | PSNR↑ | SSIM↑ |
| DPS | 0.221 | 24.28 | 0.668 | 0.225 | 24.00 | 0.664 |
| ΠGDM | 0.099 | 28.51 | 0.859 | 0.125 | 27.27 | 0.843 |
| RED-diff | 0.224 | 29.89 | 0.796 | 0.235 | 28.64 | 0.773 |
| DMPlug | 0.095 | 30.32 | 0.851 | 0.099 | 31.37 | 0.866 |
| **DMILO** | 0.076 | 32.74 | 0.897 | 0.066 | 33.05 | 0.915 |
| **DMILO-PGD** | **0.058** | **32.87** | **0.897** | **0.047** | **34.02** | **0.919** |
| **Ours vs. Best comp.** | $-0.037$ | $+2.55$ | $+0.038$ | $-0.052$ | $+2.65$ | $+0.053$ |

Table 6: (**Nonlinear IP**) **BID (Gaussian Kernel)** with additive Gaussian noise ($\sigma = 0.01$).

| | CelebA ($256 \times 256$) | | | FFHQ ($256 \times 256$) | | |
|---|---|---|---|---|---|---|
| | LPIPS↓ | PSNR↑ | SSIM↑ | LPIPS↓ | PSNR↑ | SSIM↑ |
| ILVR | 0.287 | 19.03 | 0.503 | 0.258 | 20.18 | 0.541 |
| Blind-DPS | 0.146 | 24.86 | 0.715 | 0.176 | 28.60 | 0.804 |
| DMPlug | 0.146 | 29.58 | 0.790 | 0.161 | 28.93 | 0.825 |
| **DMILO** | **0.109** | **31.20** | **0.857** | **0.156** | **29.65** | **0.854** |
| **DMILO-PGD** | 0.428 | 21.20 | 0.617 | 0.394 | 24.11 | 0.712 |
| **Ours vs. Best comp.** | -0.037 | +1.62 | +0.067 | -0.005 | +0.72 | +0.029 |

Table 7: (**Nonlinear IP**) **BID (Motion Kernel)** with additive Gaussian noise ($\sigma = 0.01$).

| | CelebA ($256 \times 256$) | | | FFHQ ($256 \times 256$) | | |
|---|---|---|---|---|---|---|
| | LPIPS↓ | PSNR↑ | SSIM↑ | LPIPS↓ | PSNR↑ | SSIM↑ |
| ILVR | 0.328 | 17.47 | 0.442 | 0.324 | 17.99 | 0.424 |
| Blind-DPS | 0.119 | 26.47 | 0.772 | 0.146 | 24.86 | 0.715 |
| DMPlug | 0.123 | 28.92 | 0.801 | 0.148 | 27.63 | 0.778 |
| **DMILO** | **0.083** | **29.84** | **0.845** | **0.078** | **30.97** | **0.888** |
| **DMILO-PGD** | 0.391 | 19.71 | 0.546 | 0.414 | 19.50 | 0.544 |
| **Ours vs. Best comp.** | -0.036 | +0.92 | +0.044 | -0.068 | +3.34 | +0.110 |

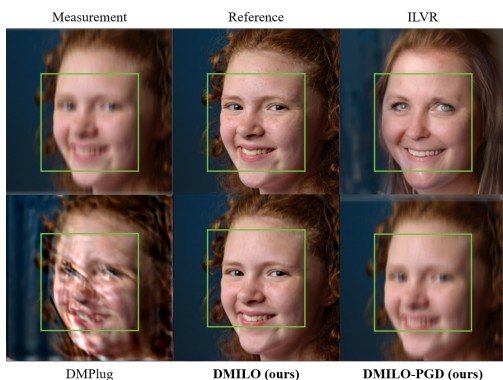

Figure 5: Visualization of sample results from our methods for BID with a motion kernel with additive Gaussian noise ($\sigma = 0.01$).

shows relatively less effectiveness. A possible reason for this is that the naive gradient update mechanism in DMILO-PGD might not be well-suited for updating the underlying blur kernel. The detailed algorithms for BID are provided in Appendix B for further reference.

## 6. Conclusion

In this paper, we presented two approaches, DMILO and DMILO-PGD, that address the memory burden and suboptimal convergence issues in DMPlug. Our theoretical analysis and numerical results illustrate the effectiveness of these proposed methods, demonstrating significant improvements over existing approaches.

## Impact Statement

Our research on inverse problems in machine learning using diffusion models makes significant contributions. The proposed DMILO and DMILO-PGD methods resolve the memory and convergence issues of DM-based solvers like DMPlug. This advancement enhances the performance of diffusion models in inverse problem applications, with broad practical use in medical imaging, compressed sensing, and remote sensing. Methodologically, it enriches the machine learning toolbox, offering theoretical insights and inspiring future algorithm development. Importantly, this work raises no ethical concerns. It aims to improve the efficiency and accuracy of data reconstruction without causing harm to individuals or society.

## Acknowledgment

This work was supported by National Natural Science Foundation of China (No.62476051, No.62176047) and Sichuan Natural Science Foundation (No.2024NSFTD0041).

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

# A. Proof of Theorem 4.4

First, we present the following basic concentration inequality for the Gaussian measurement matrix.

**Lemma A.1.** (Vempala, 2005, Lemma 1.3) *Suppose that $A \in \mathbb{R}^{m \times n}$ has i.i.d. $\mathcal{N}(0, 1/m)$ entries. For fixed $x \in \mathbb{R}^n$, we have for any $\epsilon \in (0, 1)$ that*

$$\mathbb{P}\left((1 - \epsilon)\|x\|_2^2 \leq \|Ax\|_2^2 \leq (1 + \epsilon)\|x\|_2^2\right) \geq 1 - 2e^{-\epsilon^2(1-\epsilon)m/4}. \tag{20}$$

Based on Lemma A.1 and Assumptions 4.1 and 4.2, we present the proof of Theorem 4.4 as follows.

## A.1. Proof of Theorem 4.4

From Assumption 4.2, we have

$$\log N_{\frac{\delta}{L_1}}(\mathcal{X}_2) \leq C_{g_2} k_2 \log \frac{L_1 n}{\delta}. \tag{21}$$

Additionally, using Maurey's Empirical Method (see (Daras et al., 2021, Theorem 2)), we can derive:

$$\log N_{\frac{\delta}{L_1}}(B_1^n(r)) \leq \frac{r^2 L_1^2}{\delta^2} \log(2n + 1) \leq \frac{r^2 L_1^2}{\delta^2} \log(3n). \tag{22}$$

Then, if setting $k = rL_1/\delta$, we have

$$\log N_{\frac{\delta}{L_1}}(B_1^n(r)) \leq \frac{r^2 L_1^2}{\delta^2} \log(2n + 1) \leq k^2 \log(3n). \tag{23}$$

Moreover, since $g_1$ is $L_1$-Lipschitz continuous, if $M$ is a $(\delta/L_1)$-net of $\mathcal{X}_2 + B_1^n(r)$, we have that $g_1(M)$ is a $\delta$-net of $g_1(\mathcal{X}_2 + B_1^n(r))$. Therefore, letting $\mathcal{X} = g_1(\mathcal{X}_2 + B_1^n(r))$, we have

$$\log N_\delta(\mathcal{X}) \leq C_{g_2} k_2 \log \frac{L_1 n}{\delta} + k^2 \log(3n). \tag{24}$$

Then, based on Lemma A.1 and the well-established chaining arguments in (Bora et al., 2017; Liu & Scarlett, 2020a), we have that when $m = \Omega(k_2 \log \frac{L_1 n}{\delta} + k^2 \log(3n))$, with probability $1 - e^{-\Omega(m)}$, $A$ satisfies S-REC$(\mathcal{X} - \mathcal{X}, \gamma, \delta)$. Then, we have

$$\|g_1(\hat{x}_1) - x^*\|_2 \leq \|g_1(\hat{x}_1) - g_1(\bar{x}_1)\|_2 + \|g_1(\bar{x}_1) - x^*\|_2 \tag{25}$$

$$\leq \frac{1}{\gamma}\left(\|A(g_1(\hat{x}_1) - g_1(\bar{x}_1))\|_2 + \delta\right) + \|g_1(\bar{x}_1) - x^*\|_2 \tag{26}$$

$$\leq \frac{1}{\gamma}\left(\|A(g_1(\hat{x}_1) - x^*)\|_2 + \|A(x^* - g_1(\bar{x}_1))\|_2 + \delta\right) + \|g_1(\bar{x}_1) - x^*\|_2 \tag{27}$$

$$\leq \frac{1}{\gamma}\left(2\|A(x^* - g_1(\bar{x}_1))\|_2 + \delta\right) + \|g_1(\bar{x}_1) - x^*\|_2 \tag{28}$$

$$\leq \left(1 + \frac{3}{\gamma}\right)\|x^* - g_1(\bar{x}_1)\|_2 + \frac{\delta}{\gamma}, \tag{29}$$

where (26) follows from $A$ satisfies S-REC$(\mathcal{X} - \mathcal{X}, \gamma, \delta)$, (28) follows from (18), and (29) follows from Lemma A.1 with setting $\epsilon = \frac{1}{4}$. This completes the proof.

# B. Algorithms for BID

Following the kernel estimation method proposed in DMPlug (Wang et al., 2024), we incorporate it into our framework and present the complete algorithms for our methods DMILO and DMILO-PGD for blind image deblurring tasks in Algorithms 3 and 4, respectively.

---

**Algorithm 3** DMILO for BID

---

1: **Input:** Diffusion denoising model $\mathcal{G}(\cdot) = g_1 \circ \cdots \circ g_{N-1} \circ g_N(\cdot)$, total number of sampling steps $N$, noisy observation $\boldsymbol{y}$, iteration steps $J$, Lagrange multiplier $\lambda$
2: Initialize $\boldsymbol{k}^{(0)} \sim \mathcal{N}(0, \boldsymbol{I})$, $\boldsymbol{x}_{t_N}^{(0)} \sim \mathcal{N}(\boldsymbol{0}, \boldsymbol{I})$ and $\boldsymbol{\nu}_{t_N}^{(0)} = \boldsymbol{0}$
3: **for** $i = N$ **to** $2$ **do**
4:    $\boldsymbol{x}_{t_{i-1}}^{(0)} = g_i(\boldsymbol{x}_{t_i}), \boldsymbol{\nu}_{t_{i-1}}^{(0)} = \boldsymbol{0}$
5: **end for**
6: **for** $j = 1$ **to** $J$ **do**
7:    Approximately solve $(\boldsymbol{x}_{t_1}^{(j)}, \boldsymbol{\nu}_{t_1}^{(j)}, \boldsymbol{k}^{(j)}) = \arg\min_{(\boldsymbol{x},\boldsymbol{\nu},\boldsymbol{k})} \|\boldsymbol{y} - \boldsymbol{k} * (g_1(\boldsymbol{x}) + \boldsymbol{\nu})\|_2^2 + \lambda \|\boldsymbol{\nu}\|_1$ using the Adam optimizer, initailized at $(\boldsymbol{x}_{t_1}^{(j-1)}, \boldsymbol{\nu}_{t_1}^{(j-1)}, \boldsymbol{k}^{(j-1)})$
8:    **for** $i = 2$ **to** $N$ **do**
9:       Approximately solve $(\boldsymbol{x}_{t_i}^{(j)}, \boldsymbol{\nu}_{t_i}^{(j)}) = \arg\min_{(\boldsymbol{x},\boldsymbol{\nu})} \|\boldsymbol{x}_{t_{i-1}}^{(j)} - (g_i(\boldsymbol{x}) + \boldsymbol{\nu})\|_2^2 + \lambda \|\boldsymbol{\nu}\|_1$ using the Adam optimizer, initailized at $(\boldsymbol{x}_{t_i}^{(j-1)}, \boldsymbol{\nu}_{t_i}^{(j-1)})$
10:    **end for**
11:    **for** $i = N$ **to** $1$ **do**
12:       $\boldsymbol{x}_{t_{i-1}}^{(j)} = g_i(\boldsymbol{x}_{t_i}^{(j)}) + \boldsymbol{\nu}_{t_i}^{(j)}$
13:    **end for**
14: **end for**
15: **Return** $\hat{\boldsymbol{x}} = \boldsymbol{x}_{t_0}^{(J)}$

---

## C. More Experiment Results

### C.1. Results on LSUN-bedroom Dataset

We conduct experiments on two linear inverse tasks and one nonlinear inverse task using 100 validation images randomly selected from the LSUN-Bedroom dataset. The settings of our algorithms follow the same configuration recommended in the main document. The results indicate that our algorithms DMILO and DMILO-PGD, perform well on the inpainting and nonlinear deblurring tasks, achieving the best performance across all metrics, while showing slightly lower performance on the super-resolution task.

Table 8: **Super-resolution**, **inpainting** and **nonlinear deblurring** with additive Gaussian noise ($\sigma = 0.01$) on the LSUN-bedroom dataset.

| | Super-resolution ($4\times$) | | | Inpainting (Random $70\%$) | | | Nonlinear Deblurring | | |
|---|---|---|---|---|---|---|---|---|---|
| | LPIPS↓ | PSNR↑ | SSIM↑ | LPIPS↓ | PSNR↑ | SSIM↑ | LPIPS↓ | PSNR↑ | SSIM↑ |
| DDRM | 0.200 | 27.62 | 0.820 | 0.223 | 25.53 | 0.790 | N/A | N/A | N/A |
| DPS | 0.275 | 24.03 | 0.677 | 0.124 | 29.92 | 0.863 | 0.633 | 22.40 | 0.312 |
| ΠGDM | **0.125** | 27.57 | 0.829 | 0.101 | 28.68 | 0.860 | 0.449 | 17.83 | 0.390 |
| RED-diff | 0.207 | 28.15 | **0.833** | 0.223 | 26.81 | 0.805 | 0.469 | 20.46 | 0.639 |
| DMPlug | 0.135 | 27.78 | 0.820 | 0.061 | 32.62 | 0.920 | 0.107 | 30.01 | 0.862 |
| **DMILO** | 0.167 | 27.63 | 0.790 | 0.034 | 32.62 | 0.941 | 0.061 | 31.89 | 0.919 |
| **DMILO-PGD** | 0.190 | **28.19** | 0.821 | **0.032** | **32.83** | **0.941** | **0.043** | **33.31** | **0.930** |
| **Ours vs. Best comp.** | +0.065 | +0.04 | -0.012 | -0.029 | +0.21 | +0.021 | -0.064 | +3.30 | +0.068 |

### C.2. Total Number of Sampling Steps

There exists an intuitive idea that the final sampling step alone may be sufficient to serve as the entire generator, which can be formally expressed as:

$$\mathcal{G}(\cdot) = g_1(\cdot). \tag{30}$$

To investigate this hypothesis, we conduct experiments where optimization is performed exclusively through the last timestep of the diffusion process. The results are summarized in Table 9, where "LTS" denotes methods that utilize only the last timestep for optimization.

---

**Algorithm 4** DMILO-PGD for BID

---

1: **Input:** Diffusion denoising model $\mathcal{G}(\cdot) = g_1 \circ \cdots \circ g_{N-1} \circ g_N(\cdot)$, total number of sampling steps $N$, noisy observation $\boldsymbol{y}$, iteration steps $E$, learning rate $\eta_x, \eta_k$, Lagrange multiplier $\lambda$
2: Initialize $\boldsymbol{x}_{t_0}^{(0)} = \boldsymbol{0}$, $\boldsymbol{k}^{(0)} \sim \mathcal{N}(0, \boldsymbol{I})$, $\boldsymbol{x}_{t_N}^{(0)} \sim \mathcal{N}(\boldsymbol{0}, \boldsymbol{I})$ and $\boldsymbol{\nu}_{t_N}^{(0)} = \boldsymbol{0}$
3: **for** $i = N$ **to** 2 **do**
4:     $\boldsymbol{x}_{t_{i-1}}^{(0)} = g_i(\boldsymbol{x}_{t_i}), \boldsymbol{\nu}_{t_{i-1}}^{(0)} = \boldsymbol{0}$
5: **end for**
6: **for** $e = 1$ **to** $E$ **do**
7:     $\boldsymbol{x}_{t_0}^{(e)} = \boldsymbol{x}_{t_0}^{(e-1)} - \eta_x \nabla \|\boldsymbol{y} - \boldsymbol{k}^{(e-1)} * (\boldsymbol{x}_{t_0}^{(e-1)})\|_2^2$
8:     Approximately solve $(\boldsymbol{x}_{t_1}^{(e)}, \boldsymbol{\nu}_{t_1}^{(e)}) = \arg\min_{(\boldsymbol{x},\boldsymbol{\nu})} \|\boldsymbol{k}^{(e-1)} * \boldsymbol{x}_{t_0}^{(e)} - \boldsymbol{k}^{(e-1)} * (g_1(\boldsymbol{x}) + \boldsymbol{\nu})\|_2^2 + \lambda \|\boldsymbol{\nu}\|_1$ using the Adam optimizer, initialized at $(\boldsymbol{x}_{t_1}^{(e-1)}, \boldsymbol{\nu}_{t_1}^{(e-1)})$
9:     **for** $i = 2$ **to** $N$ **do**
10:         Approximately solve $(\boldsymbol{x}_{t_i}^{(e)}, \boldsymbol{\nu}_{t_i}^{(e)}) = \arg\min_{(\boldsymbol{x},\boldsymbol{\nu})} \|\boldsymbol{x}_{t_{i-1}}^{(e)} - (g_i(\boldsymbol{x}) + \boldsymbol{\nu})\|_2^2 + \lambda \|\boldsymbol{\nu}\|_1$ using the Adam optimizer, initialized at $(\boldsymbol{x}_{t_i}^{(e-1)}, \boldsymbol{\nu}_{t_i}^{(e-1)})$
11:     **end for**
12:     **for** $i = N$ **to** 1 **do**
13:         $\boldsymbol{x}_{t_{i-1}}^{(e)} = g_i(\boldsymbol{x}_{t_i}^{(e)}) + \boldsymbol{\nu}_{t_i}^{(e)}$
14:     **end for**
15:     $\boldsymbol{k}^{(e)} = \boldsymbol{k}^{(e-1)} - \eta_k \nabla_{\boldsymbol{k}^{(e-1)}} \|\boldsymbol{y} - \boldsymbol{k}^{(e-1)} * \boldsymbol{x}_{t_0}^{(e)}\|_2^2$
16: **end for**
17: **Return** $\hat{\boldsymbol{x}} = \boldsymbol{x}_{t_0}^{(E)}$

---

Our findings indicate that optimizing solely through the last timestep remains effective to some extent, though it leads to a slight degradation in reconstruction performance compared to full multi-step optimization. We hypothesize that this performance drop is primarily due to suboptimal initialization. In principle, for the last-timestep-only approach to perform well, the initialization vector should ideally lie within the range of the composition of all preceding sampling steps. However, identifying such an initialization is nontrivial and remains an open challenge.

Further exploration of appropriate initialization strategies for the last-timestep approach will be considered as part of our future research directions.

Table 9: Experimental results for the inpainting task on 100 validation images from CelebA.

| | LPIPS↓ | PSNR↑ | SSIM↑ | FID↓ |
|---|---|---|---|---|
| DMPlug | 0.066 | 35.51 | 0.935 | 49.98 |
| DMILO | 0.025 | 36.07 | 0.951 | 19.34 |
| DMILO-LTS | 0.041 | 34.22 | 0.934 | 25.54 |
| DMILO-PGD | 0.023 | 36.42 | 0.952 | 19.08 |
| DMILO-PGD-LTS | 0.031 | 34.42 | 0.937 | 19.46 |

### C.3. Sparse Deviation

In DMILO, we introduce sparse deviations to expand the effective range of diffusion models, aiming to achieve better reconstruction performance. We hypothesize that incorporating sparse deviations can alleviate error accumulation caused by inaccurate intermediate optimization steps, thereby improving the overall quality of the reconstructed images.

To validate this hypothesis, we conduct an ablation study to examine the impact of sparse deviations on the super-resolution task using the CelebA dataset. The results are summarized in Table 10, where "w/" denotes methods that utilize sparse deviations, and "w/o" represents methods without sparse deviations.

As shown in Table 10, the inclusion of sparse deviations consistently improves performance across multiple metrics, including LPIPS, PSNR, and SSIM. These results clearly demonstrate the effectiveness of the proposed sparse deviation mechanism in enhancing reconstruction quality.

Table 10: Ablation study on the effect of sparse deviations for the super-resolution task on 100 validation images from the CelebA dataset.

|  | LPIPS↓ | PSNR↑ | SSIM↑ |
|---|---|---|---|
| DMPlug | 0.127 | 32.38 | 0.875 |
| DMILO (w/) | 0.133 | 30.81 | 0.785 |
| DMILO (w/o) | 0.202 | 29.23 | 0.699 |
| DMILO-PGD (w/) | 0.056 | 33.58 | 0.906 |
| DMILO-PGD (w/o) | 0.173 | 32.07 | 0.870 |

### C.4. Memory and Time Cost

To further validate the efficiency of our methods, we conduct a comparative experiment on the CelebA dataset for the image inpainting task to further demonstrate the time efficiency advantage of our proposed methods over DMPlug. The experiment is performed using a single NVIDIA GeForce RTX 4090 GPU, with the pretrained diffusion model having a parameter size of 357 MB. We compare the GPU memory consumption and the time required to reconstruct a single image under the optimal parameter settings for all compared methods. Additionally, we apply gradient checkpointing (Sohoni et al., 2019) to DMPlug to optimize its memory consumption, where the variant with checkpointing is referred to as "DMPlug-Ckpt" in the table.

The experimental results are presented in Table 11. The gradient checkpointing strategy effectively reduces the memory burden of DMPlug. However, this comes at the expense of increased computation time due to the overhead introduced by saving and loading intermediate gradients. In contrast, our proposed methods not only reduce memory consumption but also significantly lower the computation time compared to DMPlug, which is because our methods employ a smaller gradient graph and then lessen the burden of gradient computation.

Table 11: Memory Cost (in GB) and Time Cost (in seconds)

|  | DDRM | DPS | ΠGDM | RED-diff | DiffPIR | DAPS | DMPlug | DMPlug-Ckpt | DMILO | DMILO-PGD |
|---|---|---|---|---|---|---|---|---|---|---|
| NFE | 20 | 1000 | 50 | 50 | 20 | 500 | 15000 | 15000 | 3000 | 3000 |
| Time (s) | 1 | 40 | 2 | 1 | 1 | 43 | 925 | 1256 | 150 | 151 |
| Memory (GB) | 1.38 | 2.89 | 4.69 | 4.61 | 1.38 | 1.37 | 6.94 | 3.01 | 3.33 | 3.34 |

# D. Visualization of Experimental Results

We visualize our experimental results on two linear inverse tasks: inpainting and super-resolution and three nonlinear inverse tasks: nonlinear deblurring, blind Gaussian deblurring and blind motion deblurring tasks. All tasks are under Gaussian noise with the noise level $\sigma = 0.01$.

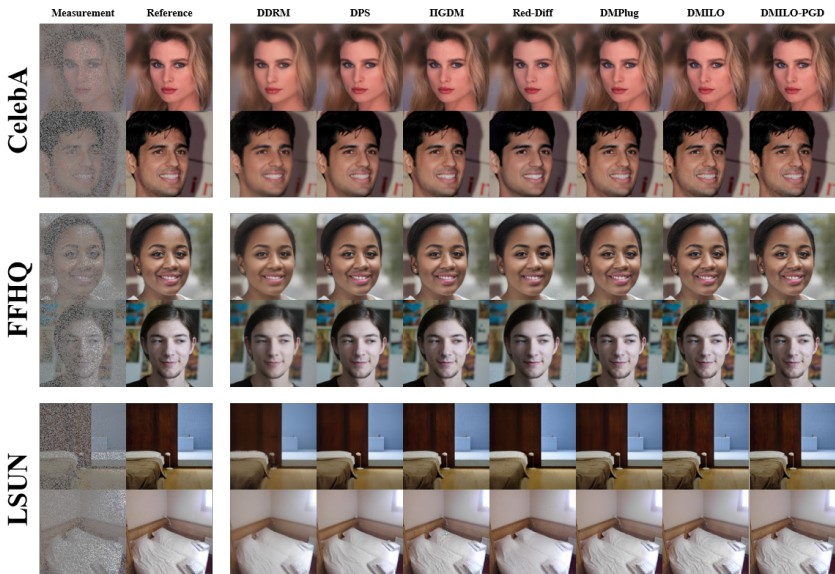

Figure 6: Visualization of sample results for inpainting with additive Gaussian noise ($\sigma = 0.01$).

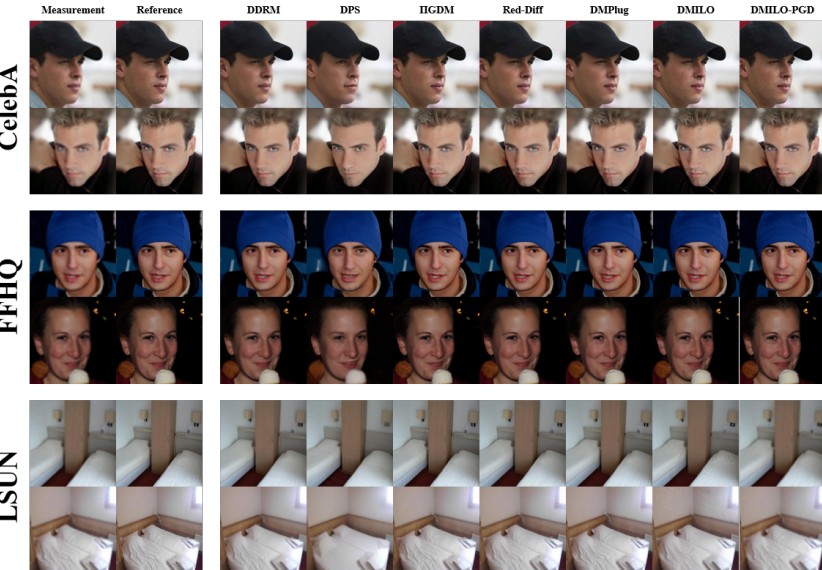

Figure 7: Visualization of sample results for $4\times$ super-resolution with additive Gaussian noise ($\sigma = 0.01$).

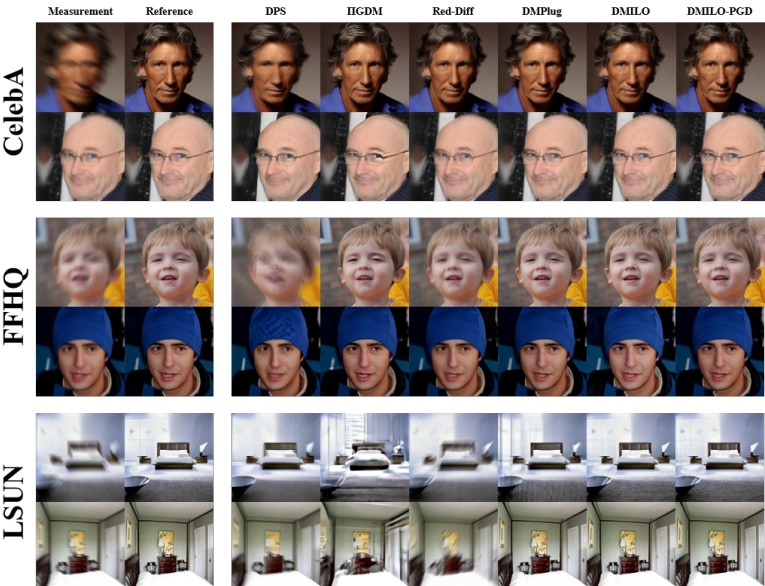

Figure 8: Visualization of sample results for nonlinear deblurring with additive Gaussian noise ($\sigma = 0.01$).

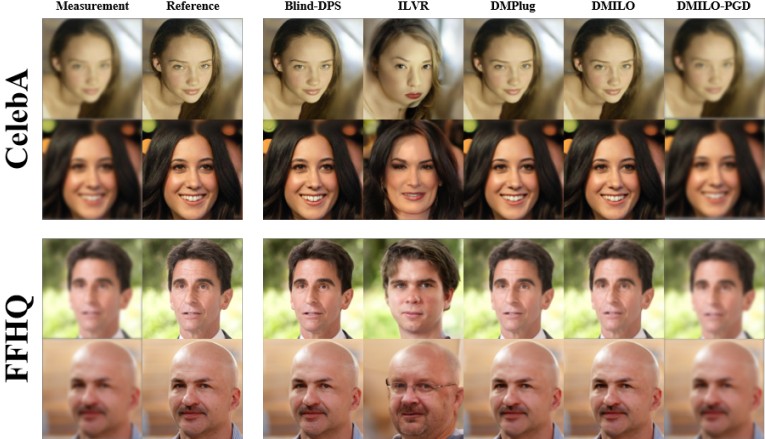

Figure 9: Visualization of sample results for BID with a Gaussian kernel with additive Gaussian noise ($\sigma = 0.01$).

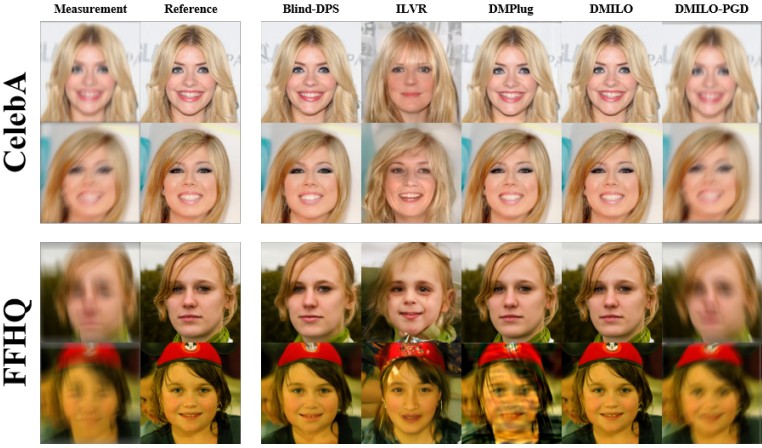

Figure 10: Visualization of sample results for BID with a motion kernel with additive Gaussian noise ($\sigma = 0.01$).

