# OpenReview forum: "Integrating Intermediate Layer Optimization and Projected Gradient Descent for Solving Inverse Problems with Diffusion Models"
_ICML.cc/2025/Conference — ICML 2025 poster_

### Official Review · Reviewer_oTx5 · 2025-02-22

**Overall Recommendation:** 3

**Summary:**

This paper proposes a novel algorithm for zero-shot inverse problem solving using diffusion models. The method build off the recent DMPlug model, which optimizes the input to conform with partial methods. The authors highlight a key insight, the optimization through the diffusion sampling process can be done more efficiently when applied to each step separately. The paper validates the result of their method, and offers another variant based on PGD.

**Claims And Evidence:**

The claim in the paper are valid but I believe some supporting evidence is missing, see method and evaluation criteria.

**Essential References Not Discussed:**

The authors do not mention the Perception-Distortion Tradeoff [1], which is key is analyzing inverse problem solutions using both distortion and perception metrics.

[1] Blau, Yochai, and Tomer Michaeli. "The perception-distortion tradeoff." Proceedings of the IEEE conference on computer vision and pattern recognition. 2018.

**Experimental Designs Or Analyses:**

The experimental design seems valid to me, withholding the concerns raised in ``Methods And Evaluation Criteria''.

**Methods And Evaluation Criteria:**

Overall, the metrics do fit the problem, yet, I believe some issue remain:
 - The metrics shown do not include image perceptual quality metrics, like FID or KID. Without such metrics, it is hard to access the realism of the images, or whether they are closer to MMSE estimations given the observed data $\mathbf{y}$. The example in Fig. 5 does in fact show blurred outputs, as typical of MMSE estimators.
 - Evaluation datasets are limited in scope. Can the method operate on general images (such as ImageNet)?
 - How does the method compare to alternative approaches in terms of NFEs? As a method that utilizes multiple derivatives of each diffusion step may be computationally prohibitive.

**Other Comments Or Suggestions:**

No other comments.

**Other Strengths And Weaknesses:**

The paper is well written and easy to follow.

**Questions For Authors:**

Could the authors explain why it is insufficient to only optimize through the last timestep of the diffusion process?

**Relation To Broader Scientific Literature:**

- The extension to DMPlug is relevant and interesting.
- While many algorithms for solving inverse problems with diffusion models exist, the optimization based ones are underrepresented in the field.

**Theoretical Claims:**

I found the theoretical analysis sufficient.

---

> ### Author Rebuttal · Authors · 2025-04-01
>
> Thanks for your recognition of this paper and the valuable comments and suggestions. Our responses to the main concerns are given as follows.
>
> (**The metrics shown do not include image perceptual quality metrics, like FID or KID. & Evaluation datasets are limited in scope. Can the method operate on general images (such as ImageNet)?**)
>
> Thanks for the comments. We perform the experiments for linear motion deblurring on the ImageNet dataset, with reporting the FID metric. Due to the time constraint of the rebuttal period, we follow DiffPIR (https://arxiv.org/abs/2305.08995) to calculate the FID on 100 validation images. The following results show that our DMILO method performs well in terms of both realism and reconstruction metrics, demonstrating its ability to effectively balance perceptual quality and distortion. We will present relevant results for more tasks and more datasets in the revised version.
>
> **Table D1: Experimental results for the linear motion deblurring task on 100 validation images from ImageNet.**
> ||LPIPS|PSNR|SSIM|FID|
> |:---:|:---:|:---:|:---:|:---:|
> |DiffPIR|0.282|24.79|0.608|115.74|
> |DMPlug|0.285|25.49|0.696|99.87|
> |DMILO|0.098|29.67|0.841|53.77|
> |DMILO-PGD|0.183|27.60|0.755|85.51|
>
> (**How does the method compare to alternative approaches in terms of NFEs? As a method that utilizes multiple derivatives of each diffusion step may be computationally prohibitive.**)
> Our methods build on DMPlug and reduce its computational overhead. For instance, in the inpainting task, our methods outperform DMPlug while using significantly fewer NFEs. Specifically, our methods need only 3,000 NFEs in total, compared to the 15,000 NFEs required by DMPlug. Even when the number of NFEs is the same, our methods are computationally more efficient. This is because our methods employ a smaller gradient graph, which lessens the burden of gradient computation. In Table D2 below, we present the computational cost of reconstructing a validation image from the CelebA dataset for different methods for inpainting using an NVIDIA RTX 4090 GPU. The results demonstrate that our methods require less computational time than DMPlug.
>
> **Table D2: Computation cost for different approaches.**
> ||DDRM|DPS|$\Pi$GDM|RED-diff|DMPlug|**DMILO**|**DMILO-PGD**|
> |:---:|:---:|:---:|:---:|:---:|:---:|:---:|:---:|
> |NFE|20|1000|50|50|15000|3000|3000|
> |Time (s)|1|40|2|1|925|150|151|
>
>
> (**The authors do not mention the Perception-Distortion Tradeoff [1], which is key is analyzing inverse problem solutions using both distortion and perception metrics.**)
> Thank you for pointing this out. In the revised version, we will cite the mentioned paper [1] for the Perception-Distortion Tradeoff, and comprehensively explain the performance of our methods in terms of both distortion and perceptual quality.
>
> (**Could the authors explain why it is insufficient to only optimize through the last timestep of the diffusion process?**)
> Thank you for the insightful question. We conduct an ablation study on only optimizing through the last timestep of the diffusion process and find that it also works, although with slightly degraded reconstruction performance (see Table D3 below). We conjecture that the degraded reconstruction performance is primarily attributed to improper initialization. Specifically, when optimizing only through the last timestep of the diffusion process, in principle, the procedure should be initialized with a vector that lies within the range of the composition of functions corresponding to all timesteps except the last one. However, identifying such an initial vector seems to be a challenging task. We plan to delve deeper into this aspect in future research.
>
> We believe this insufficiency arises because a single sampling step may introduce errors in detail, which are difficult to correct with only sparse deviations and accumulate during optimization. We will further explore this in the future study.
>
> **Table D3: Experimental results for the inpainting task on 100 validation images from CelebA.**
> ||LPIPS|PSNR|SSIM|FID|
> |:---:|:---:|:---:|:---:|:---:|
> |DMPlug|0.066|35.51|0.935|49.98|
> |DMILO|0.025|36.07|0.951|19.34|
> |DMILO-LTS|0.041|34.22|0.934|25.54|
> |DMILO-PGD|0.023|36.42|0.952|19.08|
> |DMILO-PGD-LTS|0.031|34.32|0.937|19.46|
>
> ("LTS" denotes methods that optimize only through the last timestep of the diffusion process.)

---

> > ### Comment · Reviewer_oTx5 · 2025-04-06
> >
> > I thank the authors for their rebuttal. I do not find that the concerns I have raised have been resolved by the authors' answers. l maintain the original recommendation as I lean towards accepting the paper.

---

> > > ### Author Response · Authors · 2025-04-08
> > >
> > > Thank you for the update. Although we are unsure about which specific concerns remain unresolved, we are truly grateful for your positive attitude regarding the acceptance of our paper.

---

### Official Review · Reviewer_ch18 · 2025-03-10

**Overall Recommendation:** 2

**Summary:**

This paper introduces two novel methods, DMILO and DMILO-PGD, to address computational and convergence challenges in solving inverse problems (IPs) using diffusion models (DMs).

**Claims And Evidence:**

The core claims, such as memory efficiency, and improved convergence, are not supported by experiments. Thus, the claims are not convincing.

**Essential References Not Discussed:**

No.

**Ethical Review Flag:**

Flag this paper for an ethics review.

**Experimental Designs Or Analyses:**

The paper addresses the solution of inverse problems. While both linear and nonlinear inverse problems are considered, the selection of problems is rather limited and primarily confined to natural images. It is recommended that the authors expand their scope to include inverse problems in other modalities, such as linear sparse view CT reconstruction and nonlinear metal artifact reduction in CT imaging. Such an expansion would enhance the study's credibility and practical applicability.

**Methods And Evaluation Criteria:**

Yes.

**Other Comments Or Suggestions:**

None.

**Other Strengths And Weaknesses:**

1. The abstract is overly lengthy and contains redundant information. It should be more concise and focused on the key contributions and findings. Additionally, the inclusion of citations within the abstract is unconventional and should be removed. A well-structured abstract should briefly introduce the problem, describe the proposed methods, and summarize the main results without excessive detail or references.
2. The experimental setup is limited and does not comprehensively address the claimed issues of heavy computational demands and suboptimal convergence in DM-based methods. While the paper demonstrates reduced memory usage, it lacks detailed analysis of computational efficiency  and convergence behavior. More rigorous experiments are needed to validate these claims.
3. The selected inverse problems (super-resolution, inpainting, nonlinear deblurring, and blind image deblurring) are limited in scope and lack diversity in modalities (e.g., medical imaging, audio, or 3D data). Including a broader range of tasks and modalities would strengthen the generalizability of the proposed methods and better demonstrate their applicability to real-world scenarios.
4. The proposed methods, DMILO and DMILO-PGD, exhibit noticeable performance fluctuations across different tasks. The authors should provide a detailed analysis of why these fluctuations occur and whether they are related to specific properties of the tasks.
5. The paper does not adequately discuss the limitations of the proposed methods. For example: Can the methods handle highly ill-posed problems or tasks with significant noise?

**Questions For Authors:**

None.

**Relation To Broader Scientific Literature:**

The authors proposed DMILO and DMILO-PGD, integrating ILO with and without PGD respectively.

 The authors offered an intuitive theoretical analysis of the proposed approach.

**Theoretical Claims:**

### 1. Simplified Composition
- The theorem assumes **G = g₁ ∘ g₂**, whereas DMs involve **multi-step compositions** (*g₁ ∘ g₂ ∘ ⋯ ∘ gₙ*).
  The analysis does not explicitly address whether the bound generalizes to *N > 2*, leaving open questions about scalability.

### 2. Practical Relevance of Assumptions
- The **Lipschitz continuity of g₁** may not hold strictly for real-world DMs due to non-linearities in neural networks.
  However, this is a common simplification in theoretical analyses of DMs (Chen et al., 2023; Li & Yan, 2024).

### 3. Sparse Deviation Regularization
- The theorem uses **ℓ₁-regularization** for *ν*, but the paper’s experiments employ Adam optimization without explicit guarantees of sparse recovery.  This creates a gap between theory (exact **ℓ₁** minimization) and practice (approximate optimization).

---

> ### Author Rebuttal · Authors · 2025-04-01
>
> Thanks for your useful comments and questions. Our responses to the main concerns are given as follows.
>
> (**The analysis does not explicitly address whether the bound generalizes to N > 2, leaving open questions about scalability.**)
> We follow the work for ILO (Daras et al., 2021) to set $N = 2$. We believe that this is sufficient for an intuitive theoretical analysis of the effectiveness of our methods (e.g., Reviewer ve9x found that our claims are “well-supported by extensive empirical experiments and intuitive theoretical analysis”, Reviewer D1CM found that our theoretical analysis “justifies the effectiveness of the proposed methods”, and Reviewer oTx5 found “the theoretical analysis sufficient”).
>
> (**The theorem uses ℓ₁-regularization for ν, but the paper’s experiments employ Adam optimization without explicit guarantees of sparse recovery. This creates a gap between theory (exact ℓ₁ minimization) and practice (approximate optimization).**)
> Firstly, it should be noted that even the globally optimal solutions to the $\ell_1$ minimization problem might not exhibit sparsity. Secondly, given that the objective function of the $\ell_1$ minimization problem is highly non-convex and obtaining its globally optimal solutions is not feasible, in practical applications, we employ the Adam optimizer to approximately solve the $\ell_1$ minimization problem.
>
> (**It is recommended that the authors expand their scope to include inverse problems in other modalities, such as linear sparse view CT reconstruction and nonlinear metal artifact reduction in CT imaging. The selected inverse problems (super-resolution, inpainting, nonlinear deblurring, and blind image deblurring) are limited in scope and lack diversity in modalities (e.g., medical imaging, audio, or 3D data).**)
> We agree that tasks such as sparse-view CT reconstruction and metal artifact reduction in CT imaging are of practical importance. However, prior works closely related to our study, such as DDRM, DPS, $\Pi$GDM, DiffPIR, and DMPlug, have concentrated their experiments on natural images for tasks including super-resolution, inpainting, nonlinear deblurring, and blind image deblurring. As is evident from the fact that these recent papers have been published in the topmost venues in ML or CV, and/or have been highly cited, this line of investigation has been widely accepted in the community and is a highly active area of research. We believe that extending the experiments to tasks like linear sparse-view CT reconstruction and multi-modal data such as medical imaging, audio, or 3D data is beyond the scope of the current work.
>
> (**The abstract is overly lengthy and contains redundant information. It should be more concise and focused on the key contributions and findings. Additionally, the inclusion of citations within the abstract is unconventional and should be removed.**)
> Thank you for the comment. In the revised version, we will shorten the abstract to enhance its conciseness, highlight key contributions, and remove the citation.
>
> (**The experimental setup is limited and does not comprehensively address the claimed issues of heavy computational demands and suboptimal convergence in DM-based methods.**)
> Our methods build on DMPlug and reduce its computational overhead. Due to the character limit, please refer to Table D2 in our responses to Reviewer oTx5 for an illustration. As PGD is known for its capacity to alleviate the problem of suboptimal convergence (Shah & Hegde, 2018), the effectiveness of addressing suboptimal convergence in DM-based methods can be observed from the experimental results presented in Table 2 in the main document. Specifically, for super-resolution and inpainting tasks, DMILO-PGD yields the best reconstructions, thereby demonstrating this advantage.
>
> (**The proposed methods, DMILO and DMILO-PGD, exhibit noticeable performance fluctuations across different tasks. The authors should provide a detailed analysis of why these fluctuations occur and whether they are related to specific properties of the tasks.**)
> We found that for linear deblurring and BID tasks, DMILO achieves the best reconstruction performance, whereas DMILO-PGD gives comparatively inferior results. As mentioned in Section 5.3, performance fluctuations may arise from the naive gradient update, which may not be well-suited for all tasks. Nevertheless, across most tasks, both DMILO and DMILO-PGD show competitive performance. We leave the detailed analysis of why these fluctuations occur and whether they are related to specific properties of the tasks to future work.
>
> (**The paper does not adequately discuss the limitations of the proposed methods. For example: Can the methods handle highly ill-posed problems or tasks with significant noise?**)
> We primarily follow the experimental settings of DMPlug, which does not deal with highly ill-posed problems or tasks with significant noise. We leave the research on these aspects to future work.

---

> > ### Comment · Reviewer_ch18 · 2025-04-02
> >
> > The authors have addressed my concerns by placing them in the Future Work. However, these issues are fundamental to the completeness of the current study rather than merely potential future extensions. The manuscript in its present version lacks key components necessary for acceptance. I keep my previous score.

---

> > > ### Author Response · Authors · 2025-04-03
> > >
> > > Thank you for the update. We appreciate that the evaluation of a paper's value can vary among readers. For the convenience of the whole reviewing team, regarding your comment “these issues are fundamental to the completeness of the current study rather than merely potential future extensions,” our responses are as follows:
> > >
> > > In our previous responses, we stated that: (1) Extending our work to handle tasks related to CT and multi-modal data such as medical imaging, audio, or 3D data is beyond the scope of the current work. (2) The intuitive theoretical analysis for the case $N=2$ is sufficient. (3) Handling highly ill-posed problems or tasks with significant noise, as well as conducting a detailed analysis of the performance fluctuations of DMILO and DMILO-PGD across different tasks, will be left to future work. We disagree that “these issues are fundamental to the completeness of the current study,” as detailed below.
> > >
> > > (1) **Task and modality extension**: The corresponding criticisms are not unique to our paper. They apply to most literature in this field, including DDRM (Kawar et al., 2022), DPS (Chung et al., 2022), ​ $\Pi$GDM (Song et al., 2023), DiffPIR (Zhu et al., 2023), and the most relevant work DMPlug (Wang et al., 2024). These recent papers have been published in topmost ML or CV venues and/or have high citation counts (also shown on the list below), indicating that this line of investigation has been widely accepted in the community and is a highly active area of research. Thus, we believe these extensions are not fundamental to the completeness of the current study and are better addressed in a separate dedicated work.
> > >
> > > (2) **Extension of the intuitive theoretical analysis**: We follow the nice work for ILO (Daras et al., 2021) to set $N=2$. We believe this suffices for an intuitive theoretical analysis of our methods, and other reviewers concurred.
> > >
> > > (3) **Extension to highly ill-posed problems or tasks with significant noise & Detailed analysis of the performance fluctuations**: Our experimental setup closely follows that of DMPlug (Wang et al., 2024), which was recently accepted by NeurIPS 2024 without considering highly ill-posed problems or tasks with significant noise. Additionally, through extensive experiments on various tasks (super-resolution, inpainting, linear Gaussian and motion deblurring, nonlinear deblurring, and BID) and datasets (CelebA, FFHQ, LSUN-bedroom, and ImageNet), we have shown that both DMILO and DMILO-PGD perform competitively across most tasks. These results sufficiently demonstrate the effectiveness of our proposed methods.
> > >
> > > In summary, we believe our experimental and theoretical settings are widely accepted in the active research area of using diffusion models to solve imaging inverse problems, and the issues raised by the reviewer are not fundamental to the completeness of our current study.
> > >
> > > We sincerely hope that the final editorial decision on our submission would be based on the main contributions of this particular paper, rather than on the general limitations in this broad and popular line of works.
> > >
> > > References:
> > >
> > > [1] Kawar et al. "Denoising diffusion restoration models." NeurIPS, 2022. [853 citations]
> > >
> > > [2] Chung et al. "Diffusion posterior sampling for general noisy inverse problems." ICLR, 2023. [741 citations]
> > >
> > > [3] Song et al. "Pseudoinverse-guided diffusion models for inverse problems." ICLR, 2023. [272 citations]
> > >
> > > [4] Zhu et al. "Denoising diffusion models for plug-and-play image restoration." CVPR, 2023. [198 citations]
> > >
> > > [5] Wang et al. "DMPlug: A plug-in method for solving inverse problems with diffusion models." NeurIPS, 2024. [new paper]
> > >
> > > [6] Daras et al. "Intermediate layer optimization for inverse problems using deep generative models." ICML, 2021. [102 citations]

---

### Official Review · Reviewer_D1CM · 2025-03-13

**Overall Recommendation:** 4

**Summary:**

The paper proposes DMILO and DMILO-PGD, two novel methods for solving inverse problems using diffusion models. DMILO introduces Intermediate Layer Optimization (ILO) to reduce memory burden while improving reconstruction by allowing model variations. DMILO-PGD further integrates Projected Gradient Descent (PGD) to mitigate the lack of measurement fidelity in DMILO. The authors provide a theoretical analysis under certain conditions, demonstrating the effectiveness of their methods. Experiments across multiple linear and nonlinear inverse problems show significant improvements over state-of-the-art approaches in terms of memory efficiency and reconstruction quality.

**Claims And Evidence:**

The main claims and evidences are the following:

**Claim1:** Theoretical analysis justifies the effectiveness of the proposed methods.
**Evidence1:** The authors provide a low-dimensional manifold assumption, a Set-Restricted Eigenvalue Condition (S-REC), and a theorem (Theorem 4.4) proving that the learned measurement optimum is close to the true optimum under certain conditions (Section 4).

**Claim2:** The proposed method improves over state-of-the-art methods for solving inverse problems.
**Evidence2:** Large experiments are performed over a wide range of tasks and confirm the claim of the authors.

**Essential References Not Discussed:**

https://arxiv.org/abs/2008.13751
https://arxiv.org/abs/2305.08995.

**Experimental Designs Or Analyses:**

I did check experimental designs and analyses, they are performed seriously. However, some comparisons are not extremely fair for linear IP. In particular, Table 2 should be completed with linear deblurring (Gaussian or Motion), and baselines such as DPIR https://arxiv.org/abs/2008.13751 and/or DiffPIR https://arxiv.org/abs/2305.08995, which are stronger baselines than e.g. RedDIff or DPS (in my experience).

**Methods And Evaluation Criteria:**

Methods and evaluation criteria (e.g. metrics) are appropriate and convincing.

**Other Comments Or Suggestions:**

None

**Other Strengths And Weaknesses:**

Overall, this is a good paper with convincing results. Its main strength is its theoretical side. Its main limiation is its incremental aspect as well as missing baselines that I think would make the work even more convincing.

**Strengths:**
1. The approach, although incremental, is interesting
2. The theoretical justification is rigorously justified
3. Experimental results are convincing
4. The paper is well written

**Weaknesses:**
1. This is an incremental work
2. Some important linear inverse problems are missing, e.g. motion or gaussian deblurring
3. Comparisons with other relevant methods could be included, in particular in the relatively easy case considered by the authors (i.e. deblurring with low noise level), where other baselines work particularly well (e.g. PnP, see the DPIR/DiffPIR mentionned above).

**Questions For Authors:**

- Would the authors be able to add comparisons to DPIR/DiffPIR? This is the reason for my rating, which I'd be happy to reconsider. EDIT: this has been addressed by the authors.
- I disagree with the authors in Paragraph 3.2 when they state: "A key difference from conventional PGD is that we minimize $\|\|\mathcal{A}(\mathcal{G}(x))-\mathcal{A}(x)\|\|^2$ (notations simplified) rather than $\|\|\mathcal{G}(x) - x\|\|^2$ (again, notations simplified)." I think that this sentence is not very accurate: PGD minimizes the sum of two terms (here the authors mention only one), and PGD can be used to minimize any type of function - with or without $\mathcal{A}$. Furthermore, the difference between DMILO and DMILO-PGD lies (mainly) in step 4 of the PGD version, which is the usual gradient term from PGD... If I understand what the authors mean, it is probably wrt the proximal step, which would write in a different manner (e.g. $\text{argmin} \mathcal{G}(x) + \frac{1}{2}\|x-u\|_2^2$). All in all, what the authors mean in this sentence is a bit vague and could be made more precise.
- (optional) Would the authors be able to precisely state the functional Algorithm 2 minimizes (which is not precisely (12))? If yes this would be an interesting discussion in the appendix.

**Relation To Broader Scientific Literature:**

Relation to the scientific literature is well done for comparable methods, but related works to other approaches (non diffusion, e.g. PnP algorithms despite relevant, in particular in the context of implicitly learned priors) are missing, see landmark algorithms https://arxiv.org/abs/2008.13751, https://arxiv.org/abs/2305.08995.

**Theoretical Claims:**

I did check the proof of Theorem 4.4 and did not spot any mistake.

---

> ### Author Rebuttal · Authors · 2025-04-01
>
> Thanks for your positive assessment of this paper and the valuable comments and suggestions. Our responses to the main concerns are given as follows.
>
> (**Table 2 should be completed with linear deblurring (Gaussian or Motion), and baselines such as DPIR and/or DiffPIR. Would the authors be able to add comparisons to DPIR/DiffPIR?**)
> We conduct additional evaluations on linear Gaussian and motion deblurring tasks, comparing our methods with DPIR and DiffPIR on CelebA and FFHQ. The results are shown in Tables B1-B4, which reveal that while DiffPIR demonstrates outstanding performance in terms of the perceptual quality metric LPIPS and DPIR shows superiority in image distortion metrics such as PSNR and SSIM for linear Gaussian deblurring, our DMILO method generally attains the best performance across nearly all metrics, especially for linear motion deblurring. Note that we follow DMPlug to set the kernel size to $64 \times 64$, which differs from the $61 \times 61$ kernel size initially employed in DPIR and DiffPIR. Such a difference might impact the results.
>
> In the revised version, we will further incorporate comparisons with DPIR and DiffPIR for other applicable tasks.
>
> We sincerely hope that these additional comparisons have appropriately addressed your comments, and that you can kindly consider increasing your initial rating accordingly.
>
> **Table B1: Comparisons of different methods for linear Gaussian deblurring on 100 validation images from CelebA.**
> ||LPIPS|PSNR|SSIM|
> |:---:|:---:|:---:|:---:|
> |DPS|0.109|27.65|0.752|
> |RED-diff|0.221|29.59|0.808|
> |DPIR|0.256|**31.30**|**0.861**|
> |DiffPIR|**0.092**|28.91|0.791|
> |DMPlug|0.172|29.70|0.776|
> |DMILO|**0.092**|30.89|0.816|
> |DMILO-PGD|0.157|30.74|0.811|
>
> **Table B2: Comparisons of different methods for the linear Gaussian deblurring task on 100 validation images from FFHQ.**
> ||LPIPS|PSNR|SSIM|
> |:---:|:---:|:---:|:---:|
> |DPS|0.150|25.56|0.717|
> |RED-diff|0.272|27.15|0.778|
> |DPIR|0.271|29.06|0.844|
> |DiffPIR|0.119|26.88|0.769|
> |DMPlug|0.181|28.27|0.806|
> |DMILO|**0.110**|**29.60**|**0.852**|
> |DMILO-PGD|0.176|28.65|0.799|
>
> **Table B3: Comparisons of different methods for the linear motion deblurring task on 100 validation images from CelebA.**
> ||LPIPS|PSNR|SSIM|
> |:---:|:---:|:---:|:---:|
> |DPS|0.126|26.62|0.730|
> |RED-diff|0.229|27.32|0.758|
> |DPIR|0.192|31.09|0.826|
> |DiffPIR|0.117|28.35|0.773|
> |DMPlug|0.164|30.25|0.824|
> |DMILO|**0.044**|**34.15**|**0.908**|
> |DMILO-PGD|0.067|33.41|0.884|
>
> **Table B4: Comparisons of different methods for the linear motion deblurring task on 100 validation images from FFHQ.**
> ||LPIPS|PSNR|SSIM|
> |:---:|:---:|:---:|:---:|
> |DPS|0.167|24.34|0.676|
> |RED-diff|0.272|25.40|0.730|
> |DPIR|0.181|29.67|0.820|
> |DiffPIR|0.137|26.41|0.740|
> |DMPlug|0.173|28.58|0.812|
> |DMILO|**0.044**|**33.21**|**0.909**|
> |DMILO-PGD|0.079|31.66|0.857|
>
>
> (**I disagree with the authors in Paragraph 3.2 when they state: "A key difference from conventional PGD is that…"**)
> Thank you for the detailed comment. We will remove the phrase “a key difference” and reword the sentence to enhance its precision.
>
> (**Would the authors be able to precisely state the functional Algorithm 2 minimizes (which is not precisely (12))?**)
> Thank you for the question. In (12), we use $\hat{\mathbf{x}} _ {t _ 0}$ to represent the estimated signal. As Algorithm 1 does not provide an explicit form for $\hat{\mathbf{x}} _ {t _ 0}$, we substitute the inaccessible $\mathcal{A}(\hat{\mathbf{x}} _ {t _ 0})$ with the observed vector $\mathbf{y}$. In Algorithm 2, we first calculate $\mathbf{x} _ {t _ 0}^{(e)}$ via simple gradient descent. This $\mathbf{x} _ {t _ 0}^{(e)}$ serves as an explicit approximation of $\hat{\mathbf{x}} _ {t _ 0}$. We then fix $\mathbf{x} _ {t _ 0}^{(e)}$ and use $\mathcal{A}(\mathbf{x} _ {t _ 0}^{(e)})$ in place of $\mathcal{A}(\hat{\mathbf{x}} _ {t _ 0})$ in (12).

---

> > ### Comment · Reviewer_D1CM · 2025-04-05
> >
> > I thank the authors for replying to my points, in particular for the addition of the suggested baselines DPIR and DiffPIR.
> > While the work is incremental, I do not see any reason for rejecting it and therefore have increase my rating to 4 - Accept.

---

> > > ### Author Response · Authors · 2025-04-08
> > >
> > > Thank you for the update. We are truly grateful for the improved rating and your strong endorsement.

---

### Official Review · Reviewer_ve9x · 2025-03-14

**Overall Recommendation:** 3

**Summary:**

This paper proposes a novel approach for solving inverse problems using diffusion models through an iterative intermediate layer optimization strategy (DMILO). The optimization process is enhanced by introducing sparse deviations off the manifold of the diffusion trajectory, which allows the model to generalize beyond the range of the pre-trained diffusion model. Additionally, the method is further refined using projected gradient descent (PGD) to mitigate the risk of suboptimal convergence. The method achieves superior performance on linear tasks, nonlinear tasks, and blind image deblurring tasks.

**Claims And Evidence:**

The paper makes three primary claims:

1. By replacing optimization over the entire deterministic diffusion sampling process, the proposed intermediate-layer optimization reduces memory burden.
2. By leveraging sparse deviations, the approach gains additional flexibility to recover signals outside the range of the diffusion model.
3. The proposed method achieves improved empirical performance compared to baseline approaches.

These claims are well-supported by extensive empirical experiments and intuitive theoretical analysis.

**Essential References Not Discussed:**

I did not identify any critical missing references.

**Experimental Designs Or Analyses:**

The experiment design is sound and valid.

**Methods And Evaluation Criteria:**

The method is evaluated on several linear inverse problems, nonlinear deblurring, and blind image deblurring tasks using the CelebA and FFHQ datasets. The evaluation metrics include LPIPS, PSNR, and SSIM.

**Other Comments Or Suggestions:**

Figure 1 appears inconsistent with its caption.

**Other Strengths And Weaknesses:**

Strength:

1. The paper is well-motivated, addressing key challenges such as memory burden and suboptimal convergence in prior methods like DMPlug for solving inverse problems with diffusion models.
2. The writing is clear and provides strong intuition behind each design choice.
3. The proposed method consistently improves empirical performance across selected linear and nonlinear tasks.

Weakness:

1. My main concern is the computational cost of the proposed method. As described in Algorithms 1 and 2, DMILO and DMILO-PGD require solving $J\cdot N$ and $E\cdot N$ optimization problems via backward gradient passes through a **one-step** diffusion model. In contrast, the baseline DMPlug only requires solving a single optimization problem over a **three-step** diffusion model. If the total number of function evaluations (NFEs) is fixed, does the proposed method still achieve better performance? Could the authors provide additional insights into the computational efficiency of their approach?
2. The paper states that the proposed method addresses the memory burden in DMPlug. However, an alternative strategy commonly used in practice is gradient checkpointing, which reduces memory consumption at a slight computational overhead. Did the author compare in terms of memory and computation cost by applying checkpointing to DMPlug?
3. A minor issue is that DMILO-PGD performs worse on the blind image deblurring task, though a possible reason is provided.
4. There is no ablation study on the effect of sparse deviations, which is a major component in proposed method. Could the authors provide insights into how adding sparse deviations influences performance?

**Questions For Authors:**

Please see above.

**Relation To Broader Scientific Literature:**

This work contributes to the broader field by presenting a memory-efficient and empirically stronger approach for solving inverse problems through iterative diffusion inversion over the deterministic diffusion sampling process.

**Theoretical Claims:**

A convergence bound is provided in Theorem 4.4.

---

> ### Author Rebuttal · Authors · 2025-04-01
>
> Thanks for your recognition of this paper and the valuable feedback and suggestions. Our responses to the main concerns are given as follows.
>
> (**My main concern is the computational cost of the proposed method. If the total number of function evaluations (NFEs) is fixed, does the proposed method still achieve better performance? Could the authors provide additional insights into the computational efficiency of their approach?**)
> Our methods build on DMPlug and reduce its computational overhead. For instance, in the inpainting task, our methods outperform DMPlug while using significantly fewer NFEs. Specifically, our methods need only 3,000 NFEs in total, compared to the 15,000 NFEs required by DMPlug. Even when the number of NFEs is the same, our methods are computationally more efficient. This is because our methods employ a smaller gradient graph, which lessens the burden of gradient computation. In Table A1 below, we present the computational cost of reconstructing a validation image from the CelebA dataset for different methods for inpainting using an NVIDIA RTX 4090 GPU. The results demonstrate that our methods require less computational time than DMPlug.
>
> **Table A1: Computation cost for different approaches.**
> ||DDRM|DPS|$\Pi$GDM|RED-diff|DMPlug|**DMILO**|**DMILO-PGD**|
> |:---:|:---:|:---:|:---:|:---:|:---:|:---:|:---:|
> |NFE|20|1000|50|50|15000|3000|3000|
> |Time (s)|1|40|2|1|925|150|151|
>
> (**Did the author compare in terms of memory and computation cost by applying checkpointing to DMPlug?**)
> Following your suggestion, we compare in terms of memory and computation cost by applying gradient checkpointing to DMPlug (see Table A2 below). The gradient checkpointing strategy effectively reduces the memory burden significantly. However, it simultaneously increases the computation cost because of the overhead associated with saving and loading gradients. In contrast, our methods manage to decrease both the memory and computation costs compared to DMPlug.
>
> **Table A2: Memory cost for different approaches.**
> ||DMPlug|DMPlug-Ckpt|**DMILO**|**DMILO-PGD**|
> |:---:|:---:|:---:|:---:|:---:|
> |NFE|15000|15000|3000|3000|
> |Time (s)|925|1256|150|151|
> |Memory (GB)|6.94|3.01|3.33|3.34|
>
> ("DMPlug-Ckpt" denotes DMPlug with gradient checkpointing)
>
> (**There is no ablation study on the effect of sparse deviations., which is a major component in proposed method. Could the authors provide insights into how adding sparse deviations influences performance?**)
> Thank you for your insightful comment. We perform an ablation study on the impact of sparse deviations in the super-resolution task on the CelebA dataset (see Table A3 below). The results demonstrate the effectiveness of adding sparse deviations. In our understanding, adding sparse deviations not only broadens the range of diffusion models but also alleviates error accumulation resulting from inaccurate intermediate calculations, thus leading to improved reconstruction performance.
>
> **Table A3: Ablation study on the effect of sparse deviations for the super-resolution task on 100 validation images from CelebA.**
> ||LPIPS|PSNR|SSIM|
> |:---:|:---:|:---:|:---:|
> |DMPlug|0.127|32.38|0.875|
> |DMILO (w/)|0.133|30.81|0.785|
> |DMILO (w/o)|0.202|29.23|0.699|
> |DMILO-PGD (w/)|0.056|33.58|0.906|
> |DMILO-PGD (w/o)|0.173|32.07|0.870|
>
> ("w/" denotes methods employing sparse deviations, and "w/o" denotes methods without adding sparse deviations.)
>
> (**Figure 1 appears inconsistent with its caption.**)
> Thank you for pointing this out. In the revised version, we will correct this to make Figure 1 consistent with its caption.

---

### Decision · Program_Chairs · 2025-05-01

**Decision:**

Accept (poster)

**Comment:**

This paper presents a new method for inverse problem solutions by extending the ILO method to diffusions. The optimization search allows sparse deviations from the diffusion denoising, hence extending the range of the generative model. This is a natural and interesting idea.

The authors show good results for linear inverse problems and also beyond those to blind and non-linear inverse problems.
The experimental evaluation is quite thorough across a wide range of tasks, comparisons to numerous previous methods and different datasets.

The reviewers asked for additional experiments (e.g. for linear Gaussian deblurring on CelebA) and the Authors provided additional strong results.

One reviewer was critical and asked for further experimentation in additional tasks, but the paper contains already a sufficiently extensive evaluation, in my opinion.

Overall this is a solid contribution on an important and competitive research space.